

# Comparison of GTO-ECV and adjusted MERRA-2 total ozone columns from the last 2 decades and assessment of interannual variability

**Melanie Coldewey-Egbers**[1], **Diego G. Loyola**[1], **Gordon Labow**[2,3], **and Stacey M. Frith**[2,3]

[1]German Aerospace Center (DLR), Remote Sensing Technology Institute, Oberpfaffenhofen, Germany
[2]Science Systems and Applications Inc., Lanham, Maryland, USA
[3]Atmospheric Chemistry and Dynamics Laboratory, Code 614, NASA Goddard Space Flight Center, Greenbelt, Maryland, USA

**Correspondence:** Melanie Coldewey-Egbers (melanie.coldewey-egbers@dlr.de)

**Abstract.** In this study we compare the satellite-based Global Ozone Monitoring Experiment (GOME)-type Total Ozone Essential Climate Variable (GTO-ECV) record, generated as part of the European Space Agency's Climate Change Initiative (ESA-CCI) ozone project, with the adjusted total ozone product from the Modern Era Retrospective Analysis for Research and Applications version 2 (adjusted MERRA-2) reanalysis, produced at the National Aeronautics and Space Administration (NASA) Global Modeling and Assimilation Office (GMAO). Total ozone columns and associated standard deviations show a very good agreement in terms of both spatial and temporal patterns during their 23-year overlap period from July 1995 to December 2018. The mean difference between adjusted MERRA-2 and GTO-ECV $5° \times 5°$ monthly mean total ozone columns is $-0.9 \pm 1.5\%$. A small discontinuity in the deviations is detected in October 2004, when data from the Ozone Monitoring Instrument (OMI) were ingested in the GTO-ECV and adjusted MERRA-2 data records. This induces a small overall negative drift in the differences for almost all latitude bands, which, however, does not exceed 1 % per decade. The mean difference for the period prior to October 2004 is $-0.5 \pm 1.7\%$, whereas the difference is $-1.1 \pm 1.2\%$ for the period from October 2004 to December 2018. The variability in the differences is considerably reduced in the period after 2004 due to a significant increase in data coverage and sampling. In the tropical region, the differences indicate a slight zonal variability with negative deviations over the

Atlantic, Africa, and the Indian Ocean and positive deviations over the Pacific. Ozone anomalies and the distribution of their statistical moments indicate a very high correlation among both data records as to the temporal and spatial structures. Furthermore, we evaluate the consistency of the data sets by means of an empirical orthogonal function (EOF) analysis. The interannual variability is assessed in the tropics, and both GTO-ECV and adjusted MERRA-2 exhibit a remarkable agreement with respect to the derived patterns. The first four EOFs can be attributed to different modes of interannual climate variability, and correlations with the Quasi-Biennial Oscillation (QBO), the El Niño–Southern Oscillation (ENSO) signal, and the solar cycle were found.

## 1 Introduction

The stratospheric ozone layer shields life on Earth from harmful solar ultraviolet radiation. In the late 20th century a strong decline in ozone amounts was observed that has been attributed to anthropogenic release of halocarbons into the atmosphere. In response to the dramatic loss, the Montreal Protocol (United Nations Environment Programme, 1986) was designed to protect the ozone layer by eliminating the use of ozone-depleting substances (ODSs). It was adopted in 1987, and the actions taken under the agreement have led to noticeable decreases in the concentrations of ODSs about 10 years later (Braesicke et al., 2018). With the onset of the de-

cline in ODSs a slow healing of the ozone layer is expected. However, the detection of ozone trends and its attribution to the decline in ODSs is challenging because of strong natural ozone variability, in the middle and high latitudes in particular, and complex feedback mechanisms with atmospheric dynamics and climate change (e.g., Harris et al., 2008; Weber et al., 2011). Recent studies show the first evidence of recovery in the Antarctic and the upper stratosphere in the northern middle latitudes and indicate that the decline in ODSs contributes substantially to the observed recovery (Solomon et al., 2016; Kuttippurath and Nair, 2017; Kuttippurath et al., 2018; Braesicke et al., 2018). On the other hand, no statistically significant trends over the past 2 decades could be detected in other regions or for the near-global mean total column ozone. In the lower stratosphere there is some indication of a small, nonsignificant negative trend (Ball et al., 2018; Wargan et al., 2018). Nonetheless, the overall success of the Montreal Protocol is undisputed, as the previous substantial decrease in ozone was successfully stopped, and ozone levels have remained stable, below pre-1980 values, since the turn of the century (Braesicke et al., 2018).

The aforementioned results reveal and strengthen the need for independent and consistent global long-term data records of ozone in order to identify and quantify reliable and robust trend estimates. In this regard an essential prerequisite is sufficient temporal and spatial coverage of the measurements, which in general cannot be provided by single-instrument data records. Observations from spaceborne instruments offer the required spatial coverage but, owing to their limited lifetime, merging of multiple records is necessary to achieve adequate temporal coverage. To this effect, much progress has been made during the past 2 decades and several data records have emerged and have been used for initial trend assessment (e.g., Pawson et al., 2014; Braesicke et al., 2018; Weber et al., 2018a; SPARC/IO3C/GAW, 2019). Moreover, great efforts have been made to evaluate and understand the different sources of uncertainties in the trend estimates, e.g., the trend model itself or the stability of the data records (SPARC/IO3C/GAW, 2019).

Regarding total ozone, four different merged long-term data records providing global coverage are currently available that are based on satellite sensors measuring in nadir-viewing geometry (Weber et al., 2018a; Braesicke et al., 2018). Two of them are based on the Solar Backscatter Ultraviolet (SBUV) and SBUV/2 series of satellite instruments (Frith et al., 2014, 2017; Weber et al., 2018a) and cover the period from 1979 onwards. In addition, measurements from the GOME-type (Global Ozone Monitoring Experiment) series of sensors are used to create (i) the GOME-type Total Ozone Essential Climate Variable (GTO-ECV; Coldewey-Egbers et al., 2015) and (ii) the GOME, SCIAMACHY (Scanning Imaging Absorption Spectrometer for Atmospheric Chartography), and GOME-2 (GSG; Weber et al., 2018a) data record. All of them were recently used for the analysis of decadal ozone changes and indicate very

good consistency (Braesicke et al., 2018; Weber et al., 2018a, b).

In this study we use the GTO-ECV total ozone climate data record that has been generated in the framework of the European Space Agency's Climate Change Initiative (ESA-CCI; Hollmann et al., 2013) ozone project. GTO-ECV covers the 23-year period from 1995 to 2018 and comprises measurements from GOME on board ERS-2 (second European Remote Sensing satellite), SCIAMACHY on board Envisat (Environmental Satellite), OMI/Aura (Ozone Monitoring Instrument on board Aura), and GOME-2 on board MetOp-A and MetOp-B (Meteorological Operational satellites A and B). Chiou et al. (2014) compared GTO-ECV with the SBUV-based total ozone data record provided by the National Aeronautics and Space Administration (NASA; Frith et al., 2014) and found very good agreement in zonal mean ozone columns and corresponding anomalies. In particular, the differences showed no significant trend for the 16-year overlap period from 1996 to 2011.

The focus of the present work is to compare the gridded GTO-ECV ozone product with ozone columns from the Adjusted Modern Era Retrospective Analysis for Research and Applications version 2 reanalysis data set (adjusted MERRA-2; Bosilovich et al., 2015) from July 1995 to December 2018. Reanalysis data are generated using the data assimilation technique, which allows the production of global long-term ozone fields with high spatial and temporal resolution by combining observations from satellites and/or ground-based systems with a general circulation model (Kalnay, 2003). While Wargan et al. (2017) and Davis et al. (2017) focused on the validation and analysis of zonal mean values from the reanalysis using independent satellite and ozonesonde data, as well as other reanalysis products, in this study we make use of the good spatial resolution of the ESA-CCI GTO-ECV data record and investigate the longitudinal dependence of the differences, as well as regional features. Beginning in late 2004, total ozone column data from the OMI instrument are assimilated in the MERRA-2 reanalysis. GTO-ECV also includes OMI measurements, meaning the two data sources are not completely independent. However, the OMI data assimilated by MERRA-2 are retrieved using a different algorithm than that included in GTO-ECV. To estimate the effect of the shared OMI data on our results, we analyze differences in two periods, before and after the OMI data are included in the data products.

Furthermore, we assess the impact of year-to-year changes on ozone induced by regional phenomena, e.g., the Quasi-Biennial Oscillation (QBO) or the El Niño–Southern Oscillation (ENSO) signal, and we compare ozone anomalies in terms of their distribution functions. Additionally, we carry out an empirical orthogonal function (EOF) analysis in the tropics, aiming for a detailed assessment of the consistency of both long-term data records with regard to interannual variability.

The paper is organized as follows. Section 2 contains short descriptions of the data records. In Sect. 3 we present the results of the comparison of total ozone columns, associated standard deviations, and anomalies. The interannual variability in the tropics is assessed in Sect. 4. Summary and outlook can be found in Sect. 5.

## 2 Data sets

### 2.1 GOME-type total ozone essential climate variable

The GTO-ECV total ozone climate data record has been generated in the framework of the ESA-CCI ozone project (Ozone_cci). As part of Phase I of Ozone_cci, the first version of GTO-ECV was developed (Coldewey-Egbers et al., 2015), which incorporated measurements from three nadir-viewing satellite sensors (GOME/ERS-2, SCIA-MACHY/Envisat, and GOME-2/MetOp-A) and which covered the period 1996 to 2011. During Phase II of Ozone_cci, a number of changes regarding GTO-ECV have been realized: the underlying ozone retrieval algorithm and the merging approach were improved, two more sensors were ingested (OMI/Aura and GOME-2/MetOp-B), and the data record was expanded in time (Garane et al., 2018). Version 3 of GTO-ECV now covers the 23-year period from July 1995 to December 2018.

An overview of the individual satellite sensor characteristics is provided in Table 1. All instruments are mounted on low Earth orbit platforms and measure the solar radiation reflected and scattered by the Earth's atmosphere and surface in the ultraviolet and visible wavelength range. For GTO-ECV version 3, the total ozone columns are derived using the retrieval algorithm GODFIT (GOME-type Direct FITting) version 4 (Lerot et al., 2014; Garane et al., 2018) that is applied to all sensors. Ground-based validation reveals that the mean bias between the individual level-2 ozone columns and those from reference instruments (Brewer, Dobson, and zenith-sky spectrometers) is well within $1.5 \pm 1.0\,\%$ and the drift is below 1.4 % per decade (Garane et al., 2018). In particular the inter-sensor consistency of these individual data sets is high and generally within 0.5 % in low and middle latitudes, but toward higher latitudes the data sets also present a uniform and stable behavior.

To generate the merged product, at first, the separate pixel-based (level-2) observations are converted into level-3 products per sensor, i.e., daily and monthly averages on a regular grid of $1° \times 1°$ in latitude and longitude. Then they are combined into one single cohesive record. Before merging the individual data records, corrections are applied in order to account for possible remaining inter-sensor biases and drifts. Owing to its remarkable long-term stability with respect to the ground-based reference (Garane et al., 2018), the OMI record is used as a reference basis for GTO-ECV version 3, while GOME, SCIAMACHY, GOME-2A, and GOME-2B are adjusted in terms of correction factors that depend on latitude and time (Loyola and Coldewey-Egbers, 2012). Note that in the first version of GTO-ECV (Coldewey-Egbers et al., 2015), we used GOME as the long-term reference but replaced it with OMI because of OMI's daily nearly full global coverage. Furthermore, we can take advantage of the sufficiently long overlap periods ($> 5$ years) among all sensors leading to robust estimates of the inter-sensor differences.

Finally, all available data sets are averaged into one single record that consists of monthly mean total ozone columns as well as the corresponding standard deviations and standard errors. GOME data are included only until May 2003 due to the loss of global coverage at that time (as a consequence of the permanent failure of the on-board tape recorder). SCIA-MACHY is used only until December 2004, since the validation of the corresponding level-2 data indicated some lingering issues with increasing lifetime (Garane et al., 2018). With the incorporation of OMI data in GTO-ECV in October 2004, the amount of data (see Table 1) has increased and thus the representativeness of the monthly averages is significantly improved, since OMI provides daily global coverage, along with a much finer spatial resolution compared to the predecessor sensors.

When validating GTO-ECV total ozone columns against ground-based observations, a very good agreement of 0.5 %–1.5 % peak-to-peak amplitude was found (Garane et al., 2018). In addition, the long-term drift is negligible in the Northern Hemisphere with $-0.11 \pm 0.10\,\%$ per decade for Dobson and $0.22 \pm 0.08\,\%$ per decade for Brewer collocated measurements. In the Southern Hemisphere the drift with respect to Dobson collocations is $0.23 \pm 0.09\,\%$ per decade. Hence, the target requirements of 1 %–3 % per decade, defined within the Global Climate Observing System (GCOS, 2011), are well satisfied. It has been clearly stated that the GTO-ECV data record is suitable for climate applications, such as the longer-term analyses of the ozone layer, i.e., decadal trend studies (Coldewey-Egbers et al., 2014; Weber et al., 2018a), and the evaluation of climate model simulations (Loyola et al., 2009). Both the level-2 and level-3 Climate Research Data Packages (CRDPs) are freely available via the Ozone_cci website http://cci.esa.int/ozone/ (last access: 18 March 2020).

For the comparison with adjusted MERRA-2 data, we compute $5° \times 5°$ gridded as well as $5°$ zonal monthly averages from the original $1° \times 1°$ product.

### 2.2 Adjusted MERRA-2 ozone product

The Modern Era Retrospective Analysis for Research and Applications (MERRA-2) data set was released in 2015 by NASA's Global Modeling and Assimilation Office (GMAO) (Bosilovich et al., 2015). It is produced with version 5.12.4 of the Goddard Earth Observing System (GEOS-5.12.4) atmospheric data assimilation system, whose key components

**Table 1.** Overview of individual satellite sensors included in GTO-ECV

| Instrument/platform | Period of operation | Ground pixel size | Overpass | No. of measurements | Reference |
|---|---|---|---|---|---|
| GOME/ERS-2 | June 1995–July 2011[a] | $320 \times 40\,\mathrm{km}^2$ | $10{:}30\,\mathrm{LT}$[b] | $\sim 3.5 \times 10^4\,\mathrm{d}^{-1}$ | Burrows et al. (1999) |
| SCIAMACHY/ENVISAT | August 2002–April 2012[c] | $60 \times 30\,\mathrm{km}^2$ | $10{:}00\,\mathrm{LT}$[b] | $\sim 8.0 \times 10^4\,\mathrm{d}^{-1}$ | Bovensmann et al. (1999) |
| OMI/AURA | October 2004 onwards | $13 \times 24\,\mathrm{km}^2$ | $13{:}38\,\mathrm{LT}$[b] | $\sim 1.5 \times 10^6\,\mathrm{d}^{-1}$ | Levelt et al. (2018) |
| GOME-2/MetOp-A[d] | January 2007 onwards | $40 \times 80\,\mathrm{km}^2$ | $09{:}30\,\mathrm{LT}$[b] | $\sim 2.0 \times 10^5\,\mathrm{d}^{-1}$ | Munro et al. (2016) |
| GOME-2/MetOp-B[e] | January 2013 onwards | $40 \times 80\,\mathrm{km}^2$ | $09{:}30\,\mathrm{LT}$[b] | $\sim 2.0 \times 10^5\,\mathrm{d}^{-1}$ | Munro et al. (2016) |

[a] Last month used in GTO-ECV is May 2003 (see text for more details). [b] LT is local time at the Equator. [c] Last month used in GTO-ECV is December 2004 (see text for more details). [d] In the following we refer to GOME-2 on board MetOp-A as GOME-2A. [e] In the following we refer to GOME-2 on board MetOp-B as GOME-2B.

are the GEOS-5 Atmospheric General Circulation Model (Molod et al., 2015) and the Gridpoint Statistical Interpolation (GSI) analysis scheme (Kleist et al., 2009). The assimilated data set contains data from the National Oceanic and Atmospheric Administration (NOAA) series of SBUV/2 instruments (from 1980 to September 2004), the Microwave Limb Sounder (MLS, beginning in October 2004), the Infrared Atmospheric Sounding Interferometer (IASI, starting in September 2008), the Cross-Track Infrared Sounder (on the Suomi-NPP satellite, from April 2012 onward) and Advanced Technology Microwave Sounder (on Suomi-NPP, starting in November 2011), along with total ozone observations from OMI (beginning in October 2004). By combining available measurements with global circulation model short-term forecasts, the data assimilation methodology allows the propagation of observational information by assimilated winds, resulting in global three-dimensional maps of ozone concentrations at spatial and temporal resolutions exceeding those attainable with satellite data alone. Gridded data are released at a $0.5° \times 0.625°$ latitude by longitude resolution at 72 sigma–pressure hybrid layers between the surface and 0.01 hPa. The bottom 32 layers are terrain-following while remaining model layers, from 164 to 0.01 hPa, are constant pressure surfaces.

The assimilation produces realistic global distributions of ozone in the stratosphere and upper troposphere (Stajner et al., 2008; Wargan et al., 2015; Davis et al., 2017). The column ozone values agree with NASA's Total Ozone Monitoring Spectrometer (TOMS) to $1.8 \pm 2.8\%$ in the tropics and $1.4 \pm 3.7\%$ at higher latitudes. A more detailed validation of the MERRA-2 ozone fields and parameterized ozone chemistry are discussed in Wargan et al. (2015, 2017). A principle finding was that the ozone record could not be used for trend research due to the small but discernable step functions in the data when one instrument was removed from the assimilation and/or another was added. An overview of all ozone data sources in MERRA-2 is provided in Wargan et al. (2017, their Table 1). The transition from SBUV to MLS in 2004 produced the largest of these discontinuities. We have reduced and removed these features by "normalizing" back to the complete SBUV record (1979 onwards) us-

ing the long-term ozone record found in the Merged Ozone Data Set (MOD; Frith et al., 2014).

The SBUV MOD is a time series of total column and profile ozone constructed by combining measurements from eight individual SBUV and SBUV/2 instruments. These instruments provide continual coverage from late 1978 onwards. The SBUV/2 instruments were launched into drifting orbits, such that the Equator crossing times (ECTs) drifted slowly towards the terminator. MOD includes only measurements made while the ECTs of the respective orbits were between 08:00 and 16:00 LT. After additional minimal filtering based on known instrument issues, the individual records are combined using a simple average during periods when more than one instrument is operational. In general, the differences between measurements during periods of overlap are less than the inherent instrument uncertainty (particularly for total ozone), so no external adjustments are applied. Instead, the offsets and drifts observed between instruments during overlap periods are used to estimate the uncertainty of the MOD record. Details of the total ozone MOD data set and uncertainties can be found in Frith et al. (2014).

The normalization of MERRA-2 was done by making 5° monthly zonal means for each data set (the MERRA-2 being sampled in time and space to match the individual SBUV measurements) and determining the difference between the two in Dobson units. This difference, either positive or negative, is then added to the MERRA-2 gridded data for each latitude band and month in order to keep the long-term calibration of the SBUV record and take advantage of the spatial sampling of MERRA-2. In the following we refer to the normalized MERRA-2 data set as adjusted MERRA-2.

## 3 Results and discussion

### 3.1 Zonal mean total ozone

With the normalization of the MERRA-2, the resulting monthly zonal mean adjusted MERRA-2 product is roughly equal to SBUV MOD, which is itself completely independent of GTO-ECV. The only difference between SBUV MOD and adjusted MERRA-2 is the difference in the zonal mean com-

puted at SBUV sampling compared to that computed from the full MERRA-2 sampling. However, when considering the standard deviations in the monthly zonal means and the comparisons between the spatially resolved patterns in ozone later in this work, the GTO-ECV and adjusted MERRA-2 are not completely independent because both include the OMI data after October 2004, as described in Sect. 2. Before this time, the GTO-ECV contains GOME and SCIAMACHY data, whereas SBUV/2 measurements are assimilated in MERRA-2. Thus, the GTO-ECV and adjusted MERRA-2 are completely independent prior to October 2004, but the longitudinally resolved gridded means are not completely independent after this time. However, total ozone columns from OMI are retrieved using different algorithms for GTO-ECV (GODFIT version 4) and adjusted MERRA-2 (OMI-TOMS version 8.5), respectively. Detailed information about GODFIT version 4 can be found in Lerot et al. (2014) and Rahpoe et al. (2017), and a description of OMI-TOMS version 8.5 is provided in Bhartia (2007) and McPeters et al. (2013). A main difference between both algorithms is that OMI-TOMS uses just two wavelengths – a weakly absorbing wavelength (331.2 nm) and a strong absorbing wavelength (317.5 nm) – to retrieve ozone. On the other hand, GODFIT is a direct fitting approach (Van Roozendael et al., 2012), which makes use of the high spectral resolution that OMI provides. The fitting window spans the wavelength range 325–335 nm. For an overview of the results of the geophysical validation of both data products we refer to McPeters et al. (2008, 2015) for OMI-TOMS and to Koukouli et al. (2015) and Garane et al. (2018) for GODFIT. A comparison of total ozone columns derived from GOME using GODFIT with ozone from OMI retrieved with the TOMS algorithm is presented in Lerot et al. (2010).

At first we compare 5° zonal monthly mean ozone columns and focus on the time dependence of the differences. Figure 1 shows the difference between adjusted MERRA-2 and GTO-ECV total ozone fields as a function of latitude from 1995 to 2018 (Fig. 1a) and the difference in the standard deviations that are provided with the data (Fig. 1b). The comparison of both parameters clearly shows a small change in behavior in late 2004. Therefore, for parts of our discussion we will analyze the differences separately for both time periods. The average difference in zonal mean total ozone columns is $-0.5 \pm 1.1\%$ before October 2004 and $-1.0 \pm 1.0\%$ after the introduction of OMI/Aura data in GTO-ECV and MERRA-2. In the period before October 2004 the zonal mean differences range from $-1.3 \pm 1.3\%$ in the middle latitudes of the Southern Hemisphere to $1.4 \pm 1.0\%$ in the northernmost bands. After October 2004 the differences vary from $-1.9 \pm 0.7\%$ (middle latitudes of the Southern Hemisphere) to $0.6 \pm 0.9\%$ (northernmost band). From the validation of the GTO-ECV data record (Garane et al., 2018, their Fig. 12), we know that there is a small positive bias compared to ground-based data and also with respect to MOD (which is likewise used for normalizing MERRA-2 ozone

fields). These deviations are most pronounced in the Southern Hemisphere.

Positive differences between adjusted MERRA-2 and GTO-ECV ozone columns are found in boreal summer poleward of 60° N during the entire time period and in spring in the northern part of the tropics before October 2004. In all other seasons and latitude belts differences are negative with maximum values in the Southern Hemisphere middle latitudes and under ozone hole conditions. A small number of outliers are found, mostly in high latitudes close to the polar night and before 2002, that is probably caused by sparse data coverage and, hence, nonrepresentative monthly averages in GTO-ECV or MOD. During that time period GTO-ECV exclusively consists of GOME observations and suffers from their large ground pixel sizes and global coverage that is completed only after 3 d.

The behavior of the difference in the standard deviation (Fig. 1b) also changes considerably with the introduction of OMI data in October 2004. The mean difference in the standard deviation between adjusted MERRA-2 and GTO-ECV is $-0.7 \pm 2.1$ DU and $-1.4 \pm 1.3$ DU, respectively. Prior to October 2004, adjusted MERRA-2 standard deviations are higher than GTO-ECV around 30° N and 30° S and lower elsewhere. After October 2004 adjusted MERRA-2 standard deviations are lower than GTO-ECV in all latitude bands, but differences around 30° N and 30° S are very close to zero. From 1996 to 2001 the differences indicate a drift in the middle latitudes, in particular in the Southern Hemisphere. This could be related to the significant decrease in the latitudinal coverage of NOAA-14 data due to orbital drift of this spacecraft (see Wargan et al., 2017, their Fig. 1). During that period NOAA-14 and NOAA-11 data constitute the MOD data record.

Table 2 shows the differences (annual mean as well as seasonal means) for individual latitude belts based on the entire period 1995–2018. The largest negative differences ($\sim -1.5\%$) occur year-round in the middle latitudes of the Southern Hemisphere and from September to November in the Southern Hemisphere polar latitudes ($-1.8 \pm 3.6\%$). In the Northern Hemisphere middle and high latitudes there is an apparent seasonal cycle in the differences, with positive deviations in boreal summer and negative deviations in winter. When we compute the differences in the tropics separately for the Northern Hemisphere (30–0° N) and the Southern Hemisphere (0–30° S), small positive differences are found in the north and negative differences in the south.

Next we analyze the drift in the differences in zonal mean total ozone. We fit a linear curve to the percentage difference, i.e., (adjusted MERRA-2 − GTO-ECV)/GTO-ECV, as a function of time for each 5° latitude band separately. Figure 2 shows the drift (% per decade) for three time periods used for the fit: the entire period 1995–2018 (black); the period 1995–2004, when only SBUV(-2) data are assimilated in MERRA-2, and GTO-ECV is based on GOME and SCIA-MACHY (blue); and the period 2004–2018, when OMI data

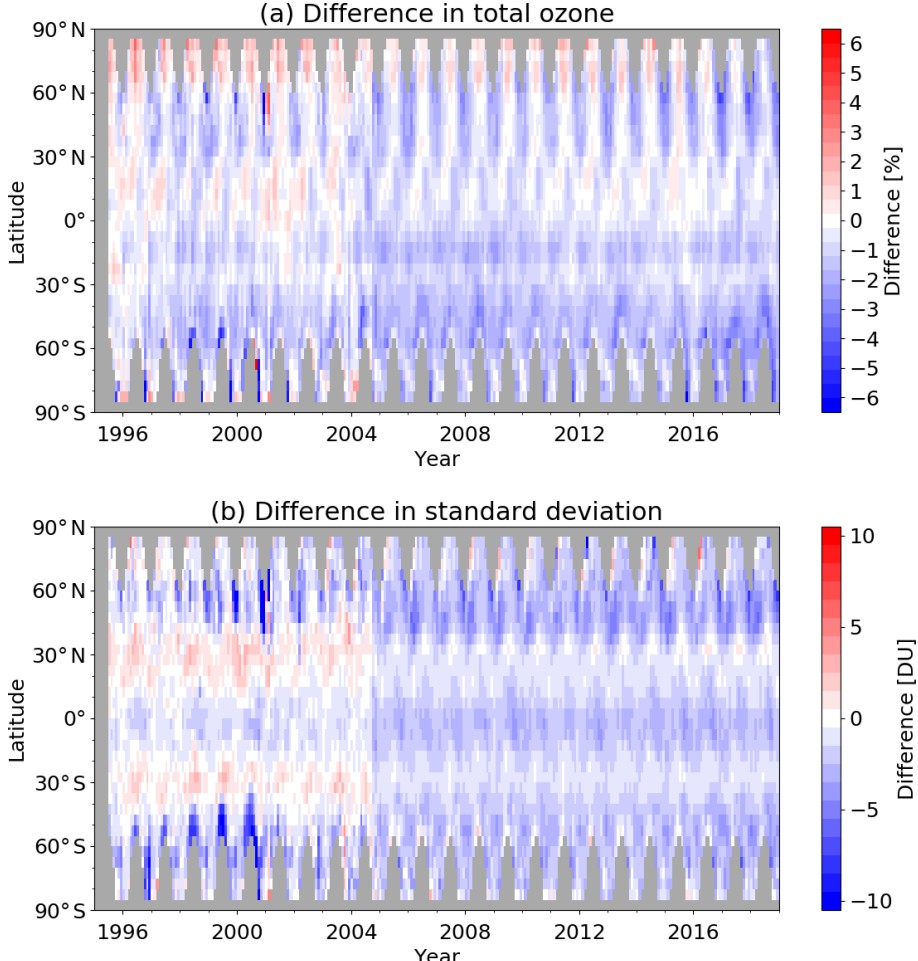

**Figure 1. (a)** Difference (%) between adjusted MERRA-2 and GTO-ECV 5° zonal monthly mean total ozone columns as a function of latitude and time from July 1995 to December 2018, computed as follows: (adjusted MERRA-2 − GTO-ECV)/GTO-ECV. **(b)** Absolute difference (DU) between adjusted MERRA and GTO-ECV 5° zonal monthly mean standard deviations provided with the products computed as follows: adjusted MERRA-2 − GTO-ECV.

**Table 2.** Difference between adjusted MERRA-2 and GTO-ECV total ozone columns for different broad latitude belts. Annual mean value and seasonal mean values for December–February (Dec–Jan–Feb), March–May (Mar–Apr–May), June–August (Jun–Jul–Aug), and September–November (Sep–Oct–Nov) are provided. Note that no data are available in the high latitudes during the polar night.

| Latitude belt | Annual mean | Dec–Jan–Feb | Mar–Apr–May | Jun–Jul–Aug | Sep–Oct–Nov |
|---|---|---|---|---|---|
| 60–90° N | $0.0 \pm 1.5\%$ | – | $0.0 \pm 1.4\%$ | $0.8 \pm 1.0\%$ | $-0.9 \pm 1.5\%$ |
| 30–60° N | $-0.9 \pm 1.5\%$ | $-1.8 \pm 1.7\%$ | $-0.8 \pm 1.3\%$ | $0.0 \pm 1.0\%$ | $-1.0 \pm 1.3\%$ |
| 30° N–30° S | $-0.6 \pm 1.0\%$ | $-0.8 \pm 1.0\%$ | $-0.5 \pm 1.0\%$ | $-0.8 \pm 1.0\%$ | $-0.5 \pm 1.1\%$ |
| 30–60° S | $-1.5 \pm 1.2\%$ | $-1.3 \pm 0.9\%$ | $-1.7 \pm 1.2\%$ | $-1.4 \pm 1.6\%$ | $-1.5 \pm 1.1\%$ |
| 60–90° S | $-1.3 \pm 2.0\%$ | $-0.7 \pm 1.3\%$ | $-1.2 \pm 1.3\%$ | – | $-1.9 \pm 2.6\%$ |

are assimilated in MERRA-2 and ingested in GTO-ECV (orange). Analysis over the entire period (black curve) indicates that the drift is slightly negative but well below 1 % per decade. Note that the uncertainty of the data records was not taken into account for this analysis. In general, the drift is stronger in the Southern Hemisphere compared to

the Northern Hemisphere, except for latitudes poleward of 60° N and 60° S. In these regions the drift is strongest (−0.85 to −0.45 % per decade) and similar for both hemispheres. When we limit the analysis to the period July 1995 to October 2004 (blue curve), the drift is also mostly negative. For the third period, 2004–2018 (orange curve), the drift is

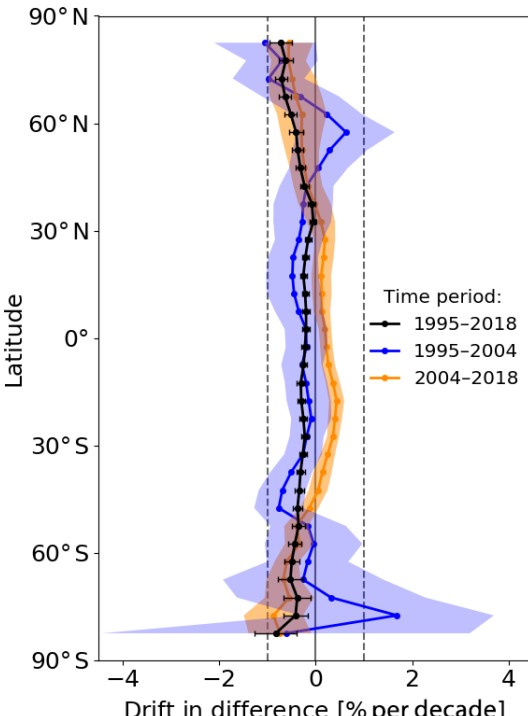

**Figure 2.** Linear drift in difference between adjusted MERRA-2 and GTO-ECV zonal monthly mean ozone columns as a function of latitude. Linear fit over (i) the entire period 1995–2018 (black), (ii) limited to the period 1995–2004 (blue), and (iii) limited to the period 2004–2018 (orange). Error bars and shading indicate the $2\sigma$ errors of the linear fit coefficients, respectively.

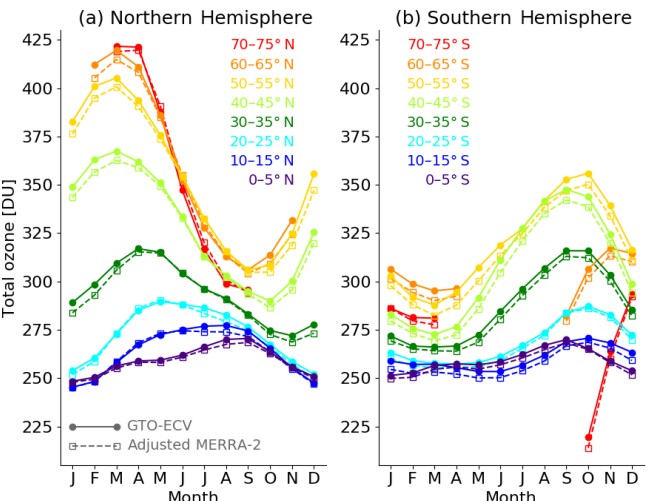

**Figure 3.** Comparison of the seasonal cycle of zonal mean total ozone columns from GTO-ECV (solid curves with filled circles) and adjusted MERRA (dashed curves with open squares) for different 5° wide latitude bands in the Northern Hemisphere **(a)** and Southern Hemisphere **(b)**. The annual cycles have been calculated using the entire period 1995–2018.

slightly positive in the tropics and in middle and high latitudes the drift is negative. This analysis reveals that the introduction of OMI data in the GTO-ECV data record leads to a slight change in the behavior of the differences, though the trend that is induced is below 1 % per decade. Nevertheless, it should be kept in mind whenever these data sets are used for trend detection in total ozone amounts.

Figure 3 shows a comparison of the annual cycle of zonal mean total ozone columns from GTO-ECV and adjusted MERRA-2 for a selection of several 5°-wide latitude bands in the Northern Hemisphere (Fig. 3a) and Southern Hemisphere (Fig. 3b). The annual cycles have been calculated from the entire overlap period 1995–2018. The curves reveal the well-known typical ozone features. A much stronger variation (peak-to-trough amplitude of ∼ 120 DU) is observed in the Northern Hemisphere compared to the Southern Hemisphere (peak-to-trough amplitude of ∼ 70 DU), with maximum values that are reached in spring in each hemisphere. An exception is the polar latitudes of the Southern Hemisphere, where extremely low values occur from September to November when the ozone hole develops. In the tropical region, the seasonal variation is less pronounced. The seasonal cycles for both data records agree quite well for all latitude bands, even for the aforementioned extreme conditions

in the high latitudes of each hemisphere. In general, adjusted MERRA-2 has a small negative bias compared to GTO-ECV (as already shown in Fig. 1) with minor exceptions, i.e., a positive bias, in the Northern Hemisphere in summer poleward of 60° N. The amplitudes of the seasonal cycles show very good agreement and differences do not exceed 2 DU.

## 3.2 Spatial patterns of differences

In this section we analyze the spatial and seasonal patterns of the gridded 5° × 5° total ozone columns, the associated standard deviations, and the corresponding differences between both data records. Figures 4 and 5 show seasonal mean total ozone columns and standard deviations for both GTO-ECV (left column) and adjusted MERRA-2 (right column), respectively. From top to bottom, the plot shows the 3-month averages for December–February (Dec–Jan–Feb), March–May (Mar–Apr–May), June–August (Jun–Jul–Aug), and September–November (Sep–Oct–Nov).

Both data records show the same (typical) spatial patterns and the same temporal evolution within a year for total ozone and standard deviation. Both parameters are low and nearly constant throughout the year in the tropical region, except for a little enhancement over the Atlantic Ocean. This enhancement is due to zonal variability in tropospheric ozone in terms of a persistent wave-one pattern (Fishman et al., 1992; Ziemke et al., 1996; Thompson et al., 2003), which maximizes near 0° longitude in the South Atlantic. The minimum occurs in the South Pacific near the date line. The amplitude of this wave pattern shows a seasonal variation with minimum values of ∼15 DU in austral autumn and maximum

values of $\sim 25$ DU in austral spring, associated with large-scale biomass burning in southern Africa and South America (e.g., Thompson et al., 2003).

Ozone amounts increase toward higher latitudes where they also indicate a clearer seasonal cycle. Maximum ozone columns are found in boreal spring in the middle latitudes of the Northern Hemisphere, whereas minimum values occur in austral spring south of 60° S. Standard deviations reach their peak values in winter and spring in the Northern Hemisphere and between September and November in the Southern Hemisphere. Furthermore, the two data records agree quite well regarding the longitudinal variability of both parameters. Winter–spring maxima in total ozone in the Northern Hemisphere are located over the Canadian Arctic and eastern Siberia, whereas a local minimum is found in the North Atlantic region (c.f. Fioletov, 2008). From September–November in the Southern Hemisphere, minimum ozone columns are found in the 0–60° W region, while high values are located in the opposite area (120–180° E). This displacement of the polar vortex toward the South Atlantic Ocean and South America is due to planetary wave activity (e.g., Ialongo et al., 2012).

Figure 6 shows the histograms of total ozone (Fig. 6a) and the standard deviations (Fig. 6c) for both $5° \times 5°$ data records. Numbers provided in the plots indicate the corresponding mean values and their $2\sigma$ standard deviations. In general, the shapes of the histograms show a very good agreement. Adjusted MERRA-2 data have a negative bias compared to GTO-ECV, except for total ozone columns in the range 250–300 DU. These values mainly occur in the tropics. For the standard deviations, adjusted MERRA-2 shows higher values in the range 10–20 DU, which generally corresponds to the subtropics (see Fig. 1). Figure 6b and d indicate the histograms of the differences in total ozone (top) and standard deviation (bottom). Because of the discontinuity in the differences that occurs in October 2004 (see Fig. 1), we plot the histograms of the differences separately for both periods, i.e., before and after that date. As already seen in Sect. 3.1 the mean bias in total ozone is $-0.5 \pm 1.7$ % in the first time period and $-1.0 \pm 1.1$ % in the second part. For the standard deviation the difference is $-0.6 \pm 3.3$ and $-1.7 \pm 1.6$ DU, respectively. For both parameters the small negative bias becomes larger in the second period, while the variance in the differences becomes smaller.

The spatial patterns of the difference in total ozone are presented in Fig. 7. Seasonal mean differences are shown that were computed as (adjusted MERRA-2 − GTO-ECV)/GTO-ECV for two different time periods: July 1995–September 2004 (left) and October 2004–December 2018 (right), respectively. The plots indicate that the differences do not solely depend on latitude but also on longitude, in particular in the tropics. Positive differences of about 0.5 %–1.0 % occur in the tropical Pacific and in the northern part of the tropical Atlantic. On the other hand, negative differences of $-1.5$ % to $-2.5$ % occur in the southern part of the tropical Atlantic and over southern Africa. Since the longitudinal structure of total ozone in the tropics is mainly determined by longitudinal variation in the troposphere (Ziemke et al., 1998), differences might be related to differences in tropospheric ozone. The pattern of the differences in total ozone indicates some correlation with the climatology of tropical tropospheric ozone (e.g., Heue et al., 2016). Maximum negative differences occur in the area of maximum tropospheric ozone amounts (South Atlantic and southern Africa) and positive differences correlate with minimum values in tropospheric ozone over the Pacific. An investigation of the zonal structure of total ozone columns from GTO-ECV and adjusted MERRA-2 yields that the wave-one pattern known from tropospheric columns is visible in the total column data, too. Locations of the maximum and the minimum are identical for both data records. However, the amplitude is slightly lower for adjusted MERRA-2 compared to GTO-ECV, which leads to the observed longitudinal pattern in the differences (Fig. 7).

In the middle latitudes of the Southern Hemisphere and in high latitudes of both hemispheres differences are more or less zonally invariant while in Northern Hemisphere middle latitudes higher spatial variability is noticed. Negative differences are found mostly over Asia, the northern Pacific, and North America, while they are less pronounced in the North Atlantic and European sector.

In general, the spatial patterns are quite similar for both time periods, except for a shift toward more negative values in the second period. In the first period the variability in the differences is stronger (cf. Fig. 6), which is probably related to the much sparser data coverage during that period. GTO-ECV is limited to GOME and SCIAMACHY (see Table 1), and the adjusted MERRA-2 data record is limited to the assimilation of SBUV/2 (see Wargan et al., 2017, their Fig. 1). GOME and SCIAMACHY provide global coverage only every 3 and 6 d, respectively. The SCIAMACHY sampling pattern is moreover determined by the alternation of limb and nadir measurements. As illustrated by Coldewey-Egbers et al. (2015, their Fig. 5) this sparse sampling may have a non-negligible, i.e., adverse impact on monthly mean ozone columns, in particular in middle latitudes during months with strong natural variability. As a consequence, average values might not be fully representative for the corresponding month and, moreover, might reflect the sampling pattern.

Figure 8 denotes the seasonal mean difference for the standard deviations. As before, we show them separately for the two periods: July 1995–September 2004 (left) and October 2004–December 2018 (right) for December–February, March–May, June–August, and September–November (from top to bottom). In contrast to total ozone differences, we compute the difference for the standard deviation as the absolute difference: adjusted MERRA-2 − GTO-ECV. All panels indicate latitudinal and longitudinal structures in the differences. During the first period, adjusted MERRA-2 standard deviations are slightly lower ($\sim 2$–2.5 DU) than GTO-ECV

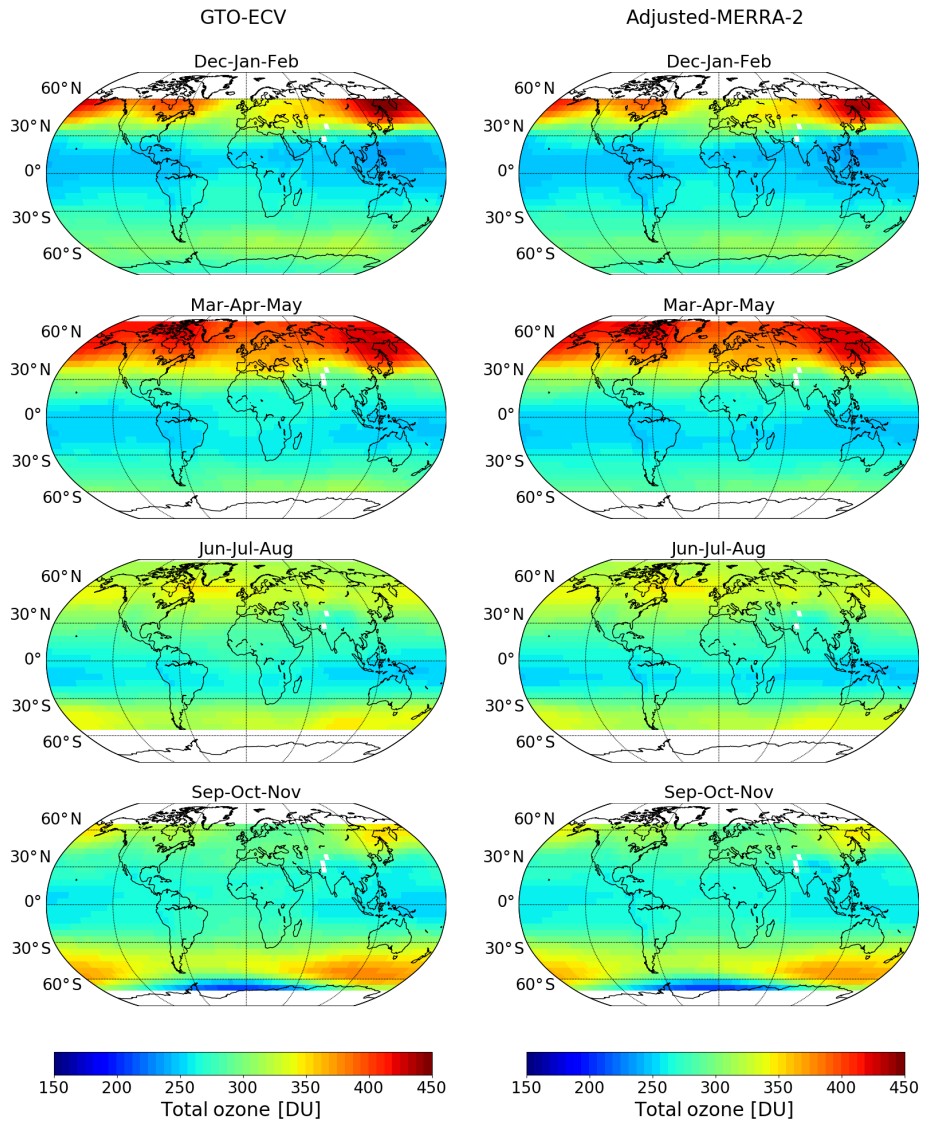

**Figure 4.** Seasonal mean total ozone columns for GTO-ECV (left column) and adjusted MERRA-2 (right column). From top to bottom: December–February (Dec–Jan–Feb), March–May (Mar–Apr–May), June–August (Jun–Jul–Aug), and September–November (Sep–Oct–Nov).

standard deviations south of 40° S, whereas in the subtropics of both hemispheres the differences are slightly positive; i.e., adjusted MERRA-2 standard deviations are higher than GTO-ECV by about 0.5 DU. In the Northern Hemisphere differences vary with season and longitude, in particular in boreal winter and spring. In the second period, the differences are negative for almost the entire globe, except for very high northern latitudes in spring and small areas in the tropics. Significant negative differences occur in the middle latitudes, most notably in the Northern Hemisphere. As for total ozone, the higher spatial variability in the differences during the first period (1995–2004) is probably related to the sparser satellite data coverage.

### 3.3 Comparison of total ozone anomalies

In addition to the differences in total ozone and standard deviation we now study ozone anomalies and their moments, i.e., standard deviation and skewness, derived from the adjusted MERRA-2 and GTO-ECV products. Anomalies are computed for each data record by subtracting the corresponding seasonal cycle over the period 1995 to 2018. Figure 9 shows the deseasonalized ozone as a function of time for seven selected 5° × 5° grid cells along 32.5° W with latitudes 72.5° N, 42.5° N, 12.5° N, 2.5° N, 12.5° S, 42.5° S, and 72.5° S, from top to bottom. Numbers in the bottom right corners denote the correlation coefficient $\rho$ between adjusted MERRA-2 and GTO-ECV anomalies, which exceeds 0.90

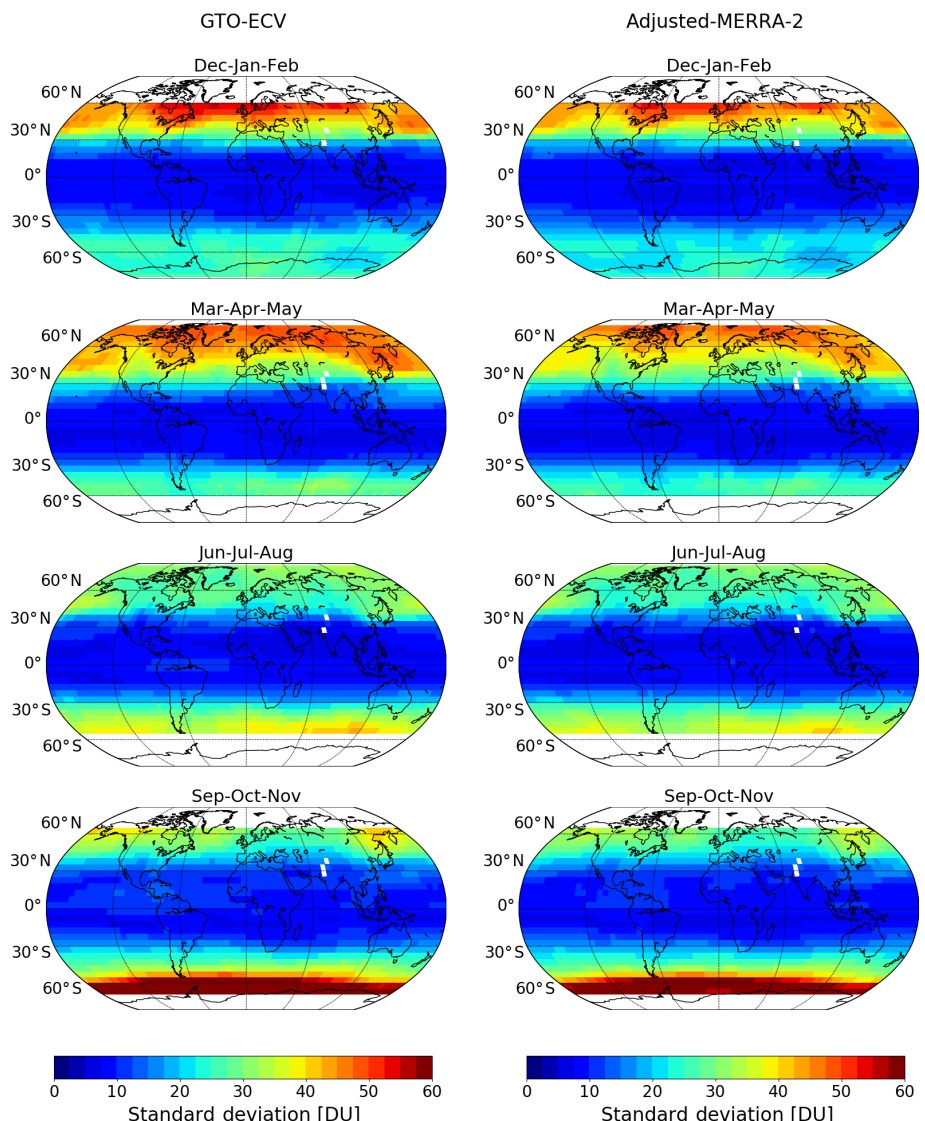

**Figure 5.** Seasonal mean standard deviations for GTO-ECV (left column) and adjusted MERRA-2 (right column). From top to bottom: December–February (Dec–Jan–Feb), March–May (Mar–Apr–May), June–August (Jun–Jul–Aug), and September–November (Sep–Oct–Nov).

in all cases except for 12.5° S. All panels indicate a very good consistency for both data records, even in high latitudes, where extreme anomalies (> 50 DU) may appear occasionally. The interannual variability in the inner tropics (2.5° N) is dominated by the QBO, and both time series agree extremely well. In this region ozone anomalies result from a QBO-induced residual circulation, i.e., ascending and descending motion (Steinbrecht et al., 2003). For instance, westerly winds lead to downward transport and an increase in total ozone. At the same time, less ozone-poor air from the lowermost layers is lifted upward. In Northern Hemisphere middle latitudes (42.5° N) two outliers in GTO-ECV in mid-2003 occur, which are most likely caused by the limited data coverage in the respective months. This period is impacted by the loss of the global coverage of GOME measurements due to the permanent failure of the on-board tape recorder. The anomalies at 12.5° S indicate a slightly worse agreement ($\rho = 0.83$), and the drift between the two data records is quite obvious here. Adjusted MERRA-2 anomalies show a positive bias compared to GTO-ECV before 2004 and a negative bias afterwards.

The correlation coefficient $\rho$ for all $36 \times 72$ (latitude $\times$ longitude) grid cells is depicted in Fig. 10. The median correlation coefficient is $\rho = 0.96$, and for 97.5 % of the grid boxes the correlation is larger than 0.90. Maximum values appear in high latitudes of both hemispheres and are affected and determined by extreme events (see Fig. 9) and in the inner tropics, which are dominated by the periodic QBO. Out-

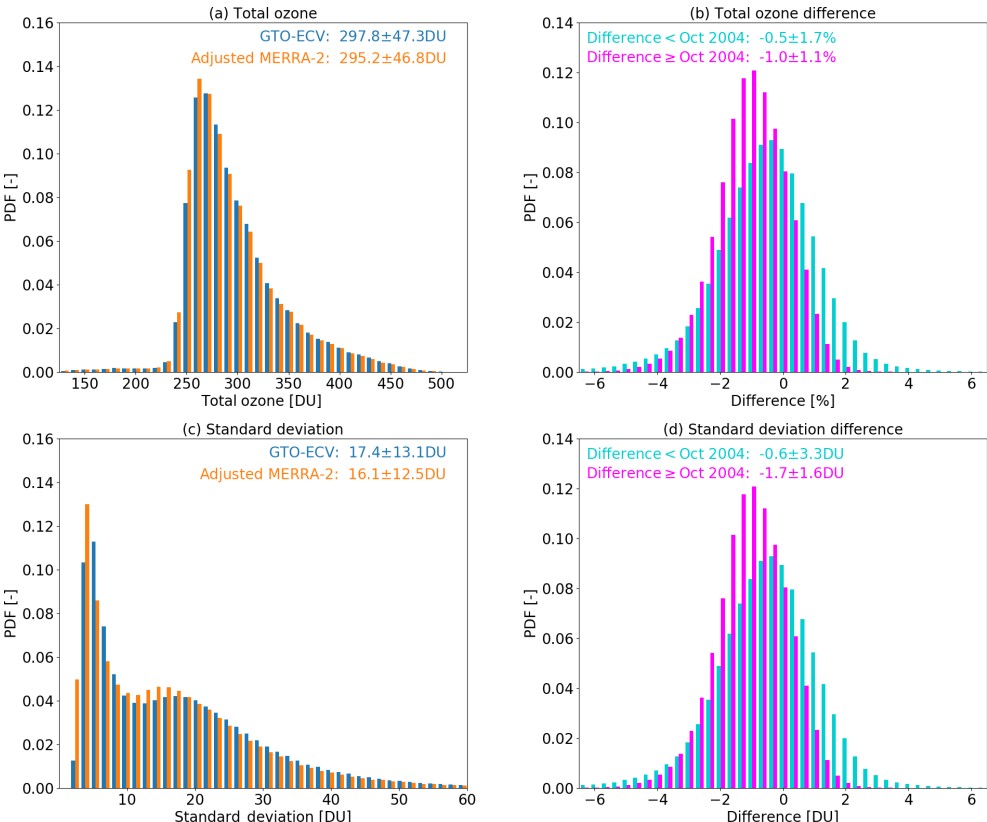

**Figure 6. (a)** Histograms of total ozone for GTO-ECV (blue) and adjusted MERRA-2 (orange). **(b)** Histogram of differences (%) in total ozone between adjusted MERRA-2 and GTO-ECV for two different time periods: before October 2004 (turquoise) and after October 2004 (magenta). **(c, d)** Corresponding histograms for standard deviations **(c)** and difference in standard deviation (DU) for two periods **(d)**.

liers are found in the region north of the Indian subcontinent (30–55° N, 70–85° E). This area suffers from regular gaps in GOME data (due to limitation of the ERS-2 tape recorder) that directly impact the quality of GTO-ECV since GOME is the only instrument during the period 1995 to 2002. In addition, lower values in the correlation ($\rho \leq 0.90$) occur in the tropical Atlantic north and south of the Equator (10–30° N and 10–30° S), which corresponds with the region of minimum interannual variability (see the next paragraph).

As a measure of the interannual variability (IAV) of ozone, we compute the standard deviation of the ozone anomalies separately for each month and compare the spatial patterns for both data records. Figure 11 shows the IAV for GTO-ECV (left) and adjusted MERRA-2 (right) for April (top) and October (bottom). Generally the IAV increases from low to high latitudes, whereas the IAV in the inner tropics (5° N–5° S) is slightly larger than for the surrounding latitude belts (5–30° N and 5–30° S). In the tropics the IAV of ozone is dominated by the QBO with influence from annual and decadal oscillations and the ENSO (Camp et al., 2003). In middle and high latitudes, the IAV is mainly governed by variations in planetary wave activity during wintertime (e.g., Fusco and Salby, 1999; Weber et al., 2011).

Therefore, the year-to-year variability reaches a maximum in high latitudes in winter and spring of each hemisphere. Furthermore, in middle and high latitudes the IAV exhibits certain longitudinal structures that are also linked to dynamic processes (Entzian and Peters, 1999; Hood et al., 1999). Figure 11 indicates an excellent agreement between both records with regard to these latitudinal and longitudinal patterns as well as to the magnitude of the IAV. The mean difference in the standard deviation of ozone anomalies between adjusted MERRA-2 and GTO-ECV is $-0.2\pm0.5$ DU or $-1.5\pm3.8$ %. That means the IAV obtained from the model-based adjusted MERRA-2 reanalysis is slightly lower than the IAV from the satellite-based record. The IAV obtained using all months reveals that the minimum variability can be found in the outer tropics in particular over the South Atlantic and southern Africa. This might be the reason for the lower correlations between GTO-ECV and adjusted MERRA-2 ozone anomalies (see Fig. 10).

Figure 12 shows a comparison of the skewness derived from the distribution of the ozone anomalies. As before, we show the results for the months April (top) and October (bottom). Following Press et al. (1992), we present only values higher than the standard deviation of the skewness, which is

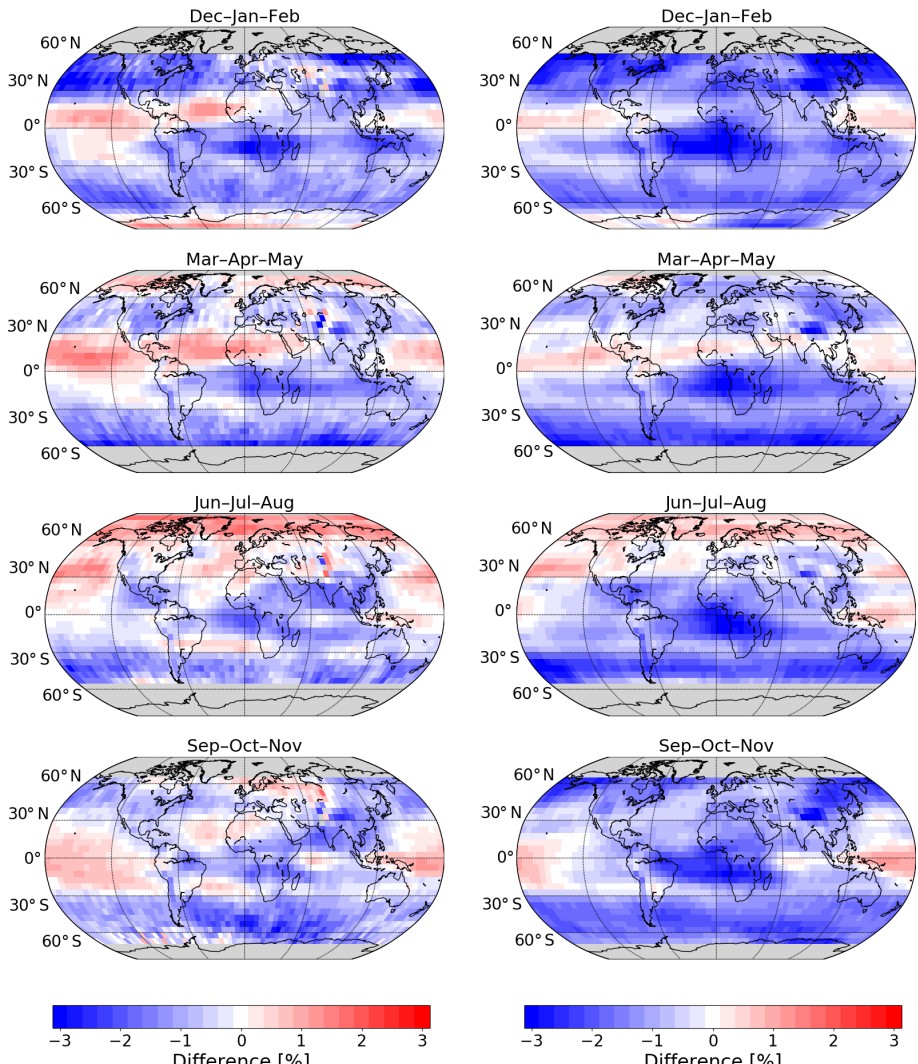

**Figure 7.** Seasonal mean percentage difference in total ozone columns between adjusted MERRA-2 and GTO-ECV. From top to bottom: December–February, March–May, June–August, and September–November. Panels on the left-hand side correspond to the period 1995–2004, and panels on the right-hand side correspond to the period 2004–2018.

defined as $\sigma_{\mathrm{skew}} = \sqrt{6/N}$. $N$ is the number of data points used for the calculation of the skewness. In case of $\sigma_{\mathrm{skew}}$ for individual months $N$ is the number of years so that the standard deviation is equal to $\sqrt{6/23} = 0.51$. Again the plots in-
5  dicate a quite good consistency of both data records in terms of the spatial patterns. In April the skewness is strongly negative in high latitudes of the Northern Hemisphere except for Greenland and northern Canada. This means that the tails of the anomaly distributions extend toward negative values. The
10  distribution of the ozone anomalies in this region is strongly impacted by severe ozone losses, i.e., significant negative anomalies, during the cold Arctic winters in the 1990s (Weber et al., 2011). On the other hand, in high latitudes of the Southern Hemisphere ozone anomalies indicate a consider-
15  able positive skewness in October (bottom panels of Fig. 12) in the region 30° W–120° E. To a large extent this is due to

the Antarctic ozone hole anomaly in 2002. In this year strong wave events and a major warming led to a split of the polar vortex and higher than normal ozone values (Stolarski et al., 2005). More noticeable positive anomalies occurred in 2010,  20 2012, and 2017, respectively. In those years the mean size of the ozone hole was much smaller ($\leq 20 \times 10^6$ km$^2$) than in other years. Negative anomalies and larger than normal ozone hole sizes occurred in 2006 (which was the severest), 2008, 2011, and 2015, although the aforementioned posi-  25 tive anomalies are more pronounced (see also Fig. 9, bottom panel) leading to the positive skewness.

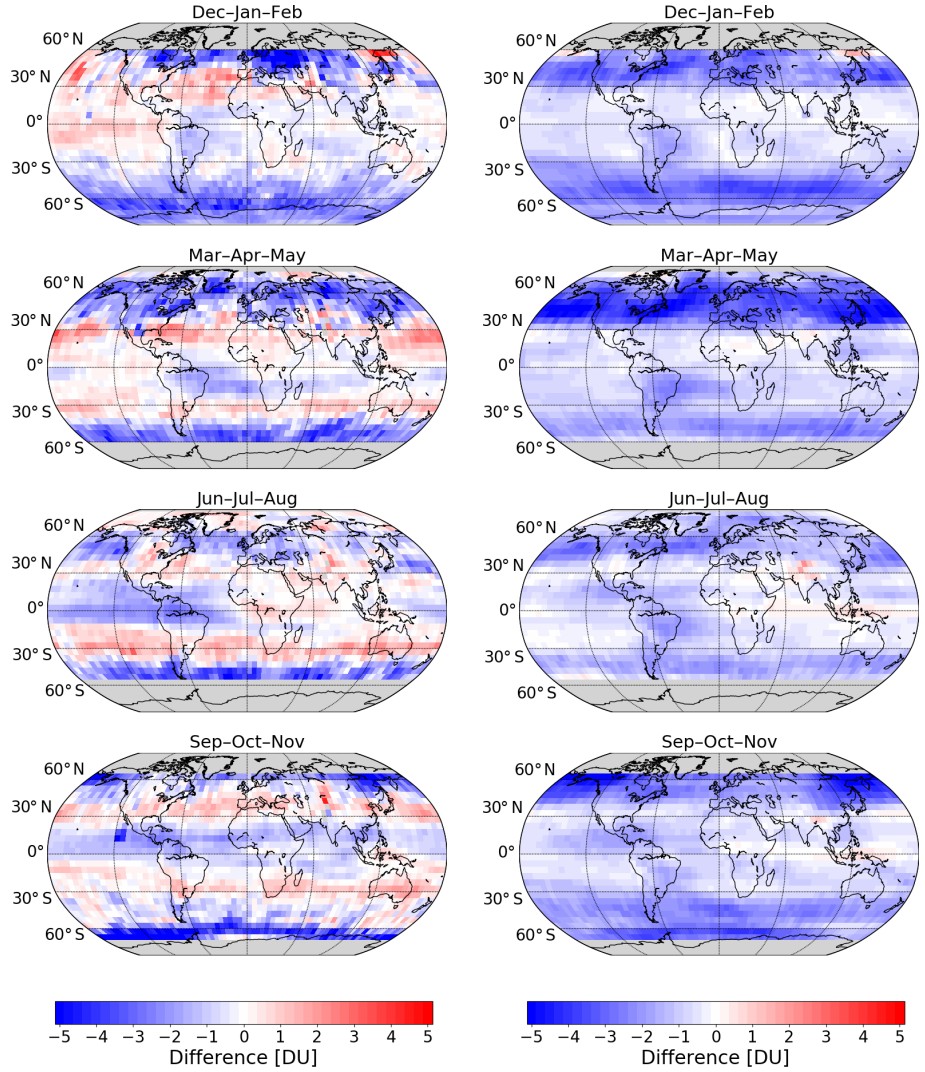

**Figure 8.** Seasonal mean absolute difference in monthly standard deviation between adjusted MERRA-2 and GTO-ECV. From top to bottom: December–February, March–May, June–August, and September–November. Panels on the left-hand side correspond to the period 1995–2004, and panels on the right-hand side correspond to the period 2004–2018.

## 4   Empirical orthogonal function analysis in the tropics

To extend the comparison of the interannual variability of ozone we carry out an empirical orthogonal function (EOF) analysis (Preisendorfer, 1988) on both data records. Similar to Camp et al. (2003) we restrict the investigation to the tropical belt from 25° N to 25° S in order to isolate and unravel the various well-known forcings (e.g., QBO or ENSO) in this region from the stronger variations in the middle latitudes. Note that essentially the EOF analysis is not based on physical principles, but the results can sometimes be interpreted as or attributed to known climate modes. The focus of this investigation is mainly the comparison of the spatial patterns and the principal component (PC) time series and is to a lesser extent dedicated to the physical interpretation of the results. The EOF analysis is performed on the detrended

and deseasonalized 5° × 5° monthly mean ozone columns presented earlier. In addition, a Savitzky–Golay smoothing filter (Savitzky and Golay, 1964) with a window length of 13 months was applied to the anomalies in order to remove higher-frequency fluctuations from the data.

For both data records the first four EOFs account for ∼92 % of the variance which can be inferred from the computed eigenvalues. Figure 13 shows the first four EOFs (from top to bottom) for GTO-ECV (left) and adjusted MERRA-2 (right) as a function of latitude and longitude. They capture 53 %, 21 %, 16 %, and 2 % of the total variance, respectively. The spatial patterns and the magnitudes agree well with the results presented in Camp et al. (2003, their Fig. 3), who analyzed (among two other data records) the MOD ozone anomalies for the period 1978 to 2000. Note that EOFs 2 and 4 are of the opposite sign compared to Camp et al. (2003).

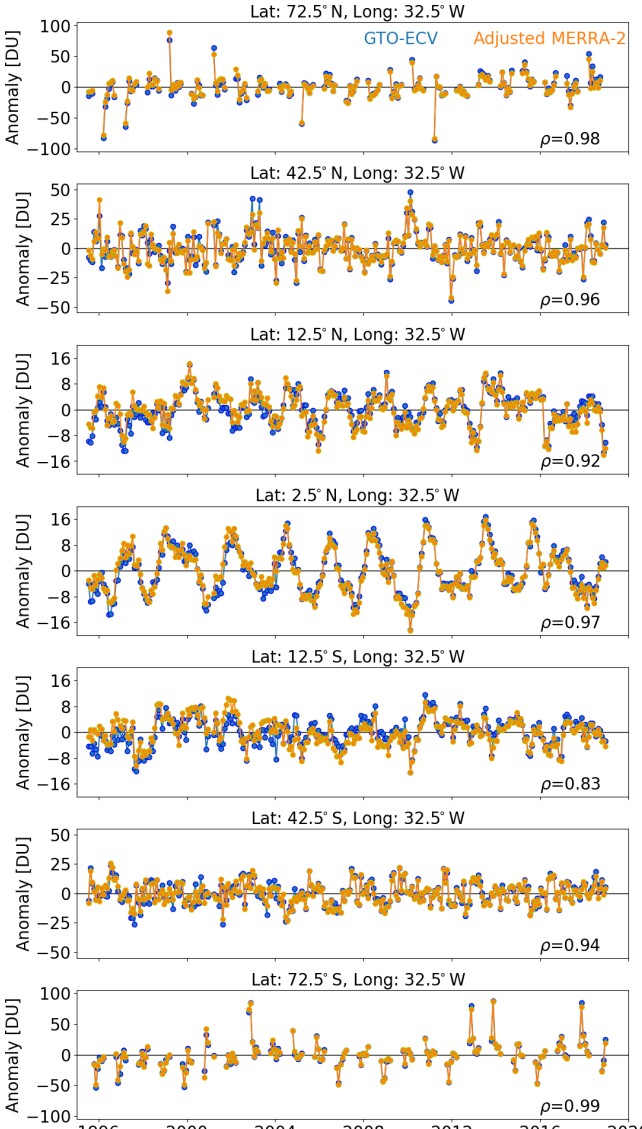

**Figure 9.** Total ozone anomalies (DU) as a function of time from 1995 to 2018 for GTO-ECV (blue) and adjusted MERRA-2 (orange) for seven selected $5° \times 5°$ grid cells along 32.5° W with latitudes from top to bottom: 72.5° N, 42.5° N, 12.5° N, 2.5° N, 12.5° S, 42.5° S, and 72.5° S. Numbers in the bottom right corner of each panel denote the correlation coefficient $\rho$ between adjusted MERRA-2 and GTO-ECV anomalies.

However, generally the signs of the eigenvectors are arbitrary and a physical interpretation will become possible by looking at EOFs and PC time series together. The EOFs of GTO-ECV and adjusted MERRA-2 show a quite good con-
5 sistency regarding the spatial structures and the range. The associated PC time series and Fourier spectra are given in Figs. 14 and 15, respectively.

The first EOFs (Fig. 13a and b) indicate zonal invariance and symmetry with respect to the Equator. EOFs are maxi-
10 mum ($\sim 7$ DU) at the Equator and the sign switches at about

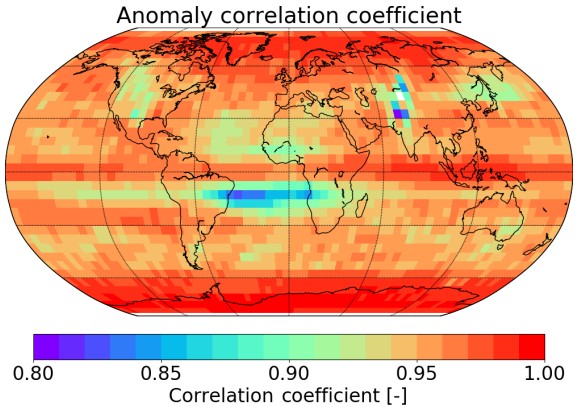

**Figure 10.** Correlation coefficient between adjusted MERRA-2 and GTO-ECV ozone anomalies.

15° N and 15° S. Minimum values are $\sim 3.3$ DU. The PC time series (Fig. 14a) indicates an amplitude of around 2, which means that the maximum peak-to-peak amplitude of ozone at the Equator is about 28 DU. PCs for both GTO-ECV and adjusted MERRA-2 agree quite well, and their 15 correlation coefficient is high ($\rho_1 = 0.99$). Figure 15a indicates a very dominant peak at a period of 28 months which is consistent with the mean period of the QBO (Baldwin et al., 2001). Therefore, in addition to the PCs, the green curve in Fig. 14a denotes the QBO index at 30 hPa (avail- 20 able at https://www.cpc.ncep.noaa.gov/data/indices/qbo.u30. index, last access: 18 March 2020), and we find a good correlation between the PCs and the QBO index ($\rho_2 = 0.81$ for GTO-ECV and $\rho_3 = 0.77$ for adjusted MERRA-2).

The second EOFs for GTO-ECV and adjusted MERRA-2 25 (Fig. 13c and d) are almost entirely positive. Only a small region between 60 and 150° E indicates negative values ($\sim -1.2$ DU). Maximum values of about 5.3 DU are found south of the Equator. The associated PCs are shown in Fig. 14b, and, as for the first EOF, they indicate a good correlation 30 ($\rho_1 = 0.94$) between GTO-ECV and adjusted MERRA-2. The Fourier spectra for the second PCs (Fig. 15b) show a dominant peak at 138 months ($\approx 11$ years) but also at $\sim 21$ and $\sim 40$ months. Figure 14b reveals a moderate correlation of the PCs with the solar cycle index (green curve; avail- 35 able at ftp://ftp.seismo.nrcan.gc.ca/spaceweather/solar_flux/ monthly_averages/solflux_monthly_average.txt, last access: 25 March 2020).

The values of the third EOFs (Fig. 13e and f) range from $-1.7$ DU ($-2.2$ DU in case of adjusted MERRA-2) in the 40 south to 4.3 DU toward the northern boundary. The change of sign occurs roughly at the Equator, and the patterns are more or less zonally invariant. Again the agreement between both data records is good. The associated Fourier spectra (Fig. 15c) indicate a strong peak at 21 months. According 45 to Tung and Yang (1994) and Camp et al. (2003) this period is a result of the interaction between the QBO and the

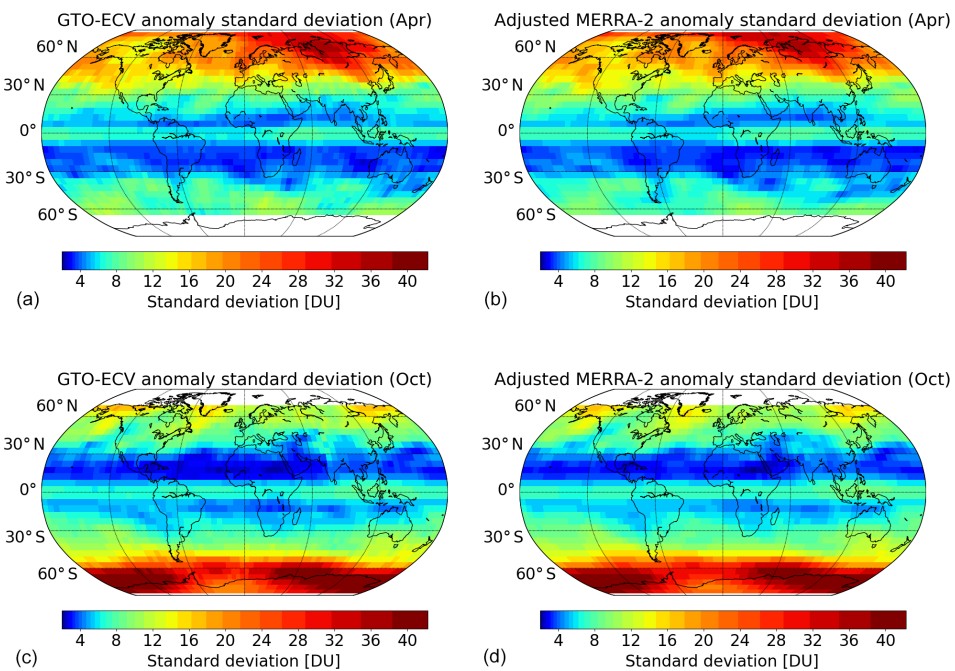

**Figure 11.** Standard deviation of ozone anomalies (DU) for GTO-ECV **(a, c)** and adjusted MERRA-2 **(b, d)** data records for April **(a, b)** and October **(c, d)**. Note the nonlinear color scale.

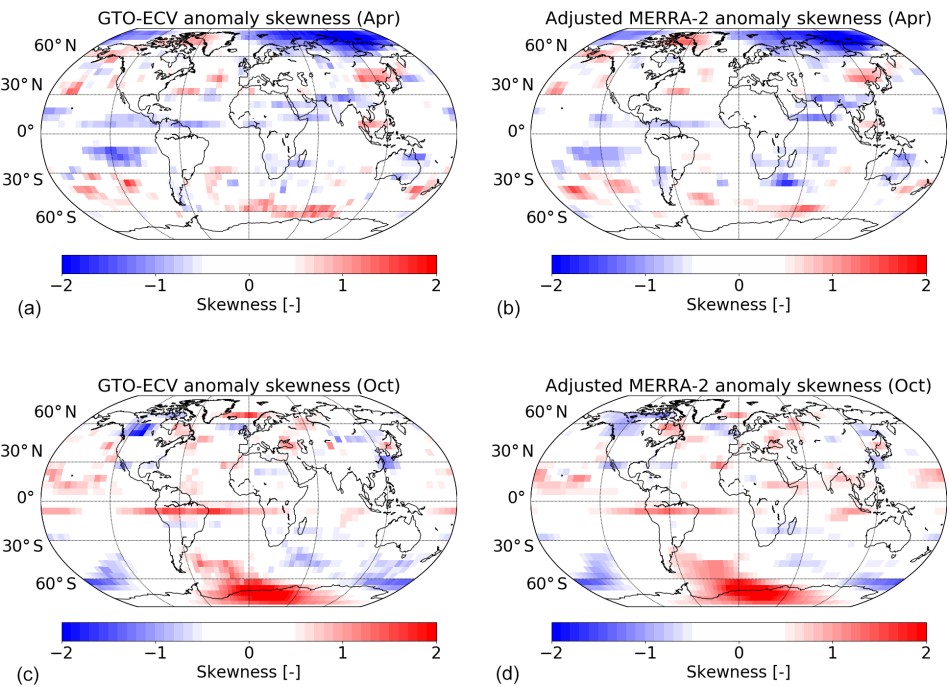

**Figure 12.** Skewness of the ozone anomaly distributions for GTO-ECV **(a, c)** and adjusted MERRA-2 **(b, d)** data records for April **(a, b)** and October **(c, d)**. Only values exceeding the standard deviation of the skewness ($\sigma_{\mathrm{skew}} = 0.51$) are presented.

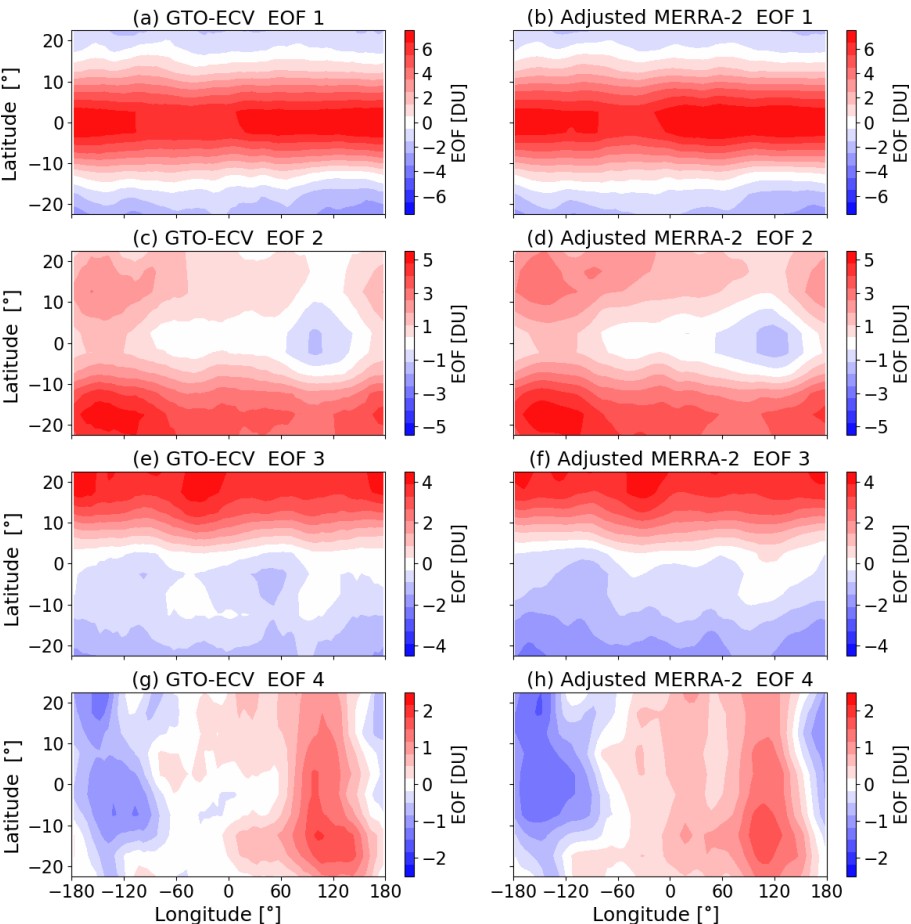

**Figure 13.** Spatial patterns for the first four EOFs (from top to bottom) in the tropics from 25° N to 25° S: **(a, c, e, g)** GTO-ECV and **(b, d, f, h)** adjusted MERRA-2.

annual cycle. The so-called QBO-annual beat frequency is the difference between the annual frequency (1/12 month) and the frequency of the QBO (1/28 month). The correlation between the PCs for GTO-ECV and adjusted MERRA-2 is quite high ($\rho_1 = 0.98$).

The spatial patterns of the fourth EOFs are presented in Fig. 13g and h, and the corresponding PCs and Fourier spectra are shown in the Figs. 14d and 15d, respectively. The EOFs show a clear zonal structure with maximum values ($\sim 1.9$ DU) in the eastern Indian Ocean and over Indonesia and the western Pacific (60–180° E) and minimum values ($\sim -1.5$ DU) over the central and eastern Pacific (90–180° W). The sign switches somewhat west of the dateline, and the maxima are found slightly south of the Equator. The PCs for GTO-ECV and adjusted MERRA-2 show an excellent agreement ($\rho_1 = 0.95$). Figure 15d indicates discrete peaks at 18 and 28 months and a broader peak for periods greater than 40 months. In contrast to the first three PCs, the dominant peaks for GTO-ECV and adjusted MERRA-2 do not agree for this PC. The dominant period for GTO-ECV is 18 months, whereas for adjusted MERRA-2 it is the decadal

signal. Wang et al. (2011) found a distinct peak at 17 months and suggested that this could be a beat frequency between ENSO and the annual cycle. Additionally, a comparison with the Multivariate ENSO Index (MEI, https://www.esrl.noaa.gov/psd/enso/mei/, last access: 18 March 2020) time series indicates a considerable correlation ($\rho \approx 0.60$) with this climate mode. In particular, the strong El Niño events in 1997 and 2015 are in accordance with positive peaks in the PC time series (Fig. 14d). We assume that the rather irregular periodicity of ENSO events ($\sim 3$–7 years) is responsible for the broad peak with substantial power for periods greater than 40 months.

The investigation of the EOFs and associated PC time series inferred from GTO-ECV and adjusted MERRA-2 total ozone anomalies in the tropics has demonstrated an excellent agreement among the two long-term data records in terms of both spatial and temporal patterns. PC time series indicate a high correlation and also the derived spectral features are very consistent. Furthermore, the extracted structures can be attributed to different modes of interannual dynamically induced climate variability. As shown in Fig. 14 and discussed

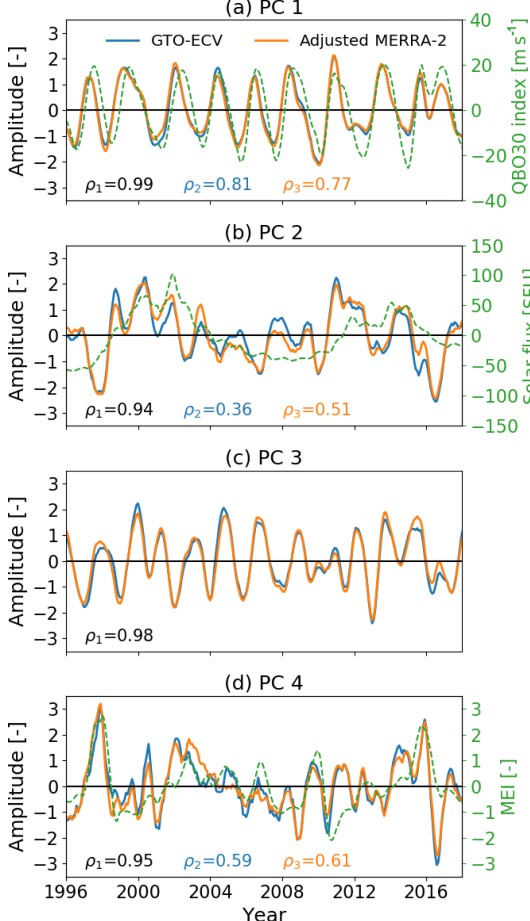

**Figure 14.** Principal component (PC) time series for the first four EOFs (from top to bottom) for GTO-ECV (blue) and adjusted MERRA-2 (orange). The green curves denote appropriate climatic indices: **(a)** QBO at 30 hPa, **(b)** solar flux at 10.7 cm, and **(d)** the Multivariate ENSO Index (MEI). All indices were detrended, and a Savitzky–Golay smoothing filter with a window length of 13 months was applied. The solar flux is given in solar flux units (SFU), defined as $1\,\mathrm{SFU} = 10^{-22}\,\mathrm{W\,m^{-2}\,Hz^{-1}}$. The numbers provided in the bottom part of the plots indicate the correlation coefficients between GTO-ECV and adjusted MERRA-2 PCs ($\rho_1$, black), between the GTO-ECV PC and the selected proxy ($\rho_2$, blue), and the adjusted MERRA-2 PC and the selected proxy ($\rho_3$, orange), respectively. For PC3 **(c)** no proxy is shown (see text for more details).

in, e.g., Tung and Yang (1994), Camp et al. (2003), or Jiang et al. (2004), to a large extent the QBO, the solar cycle, and ENSO induce year-to-year changes in ozone.

Regarding the QBO at 30 hPa, an in-phase relation be-
5 tween the mean zonal wind and total ozone was observed for the inner tropical belt ($\pm 15°$), i.e., high ozone during westerly winds and low ozone during easterly winds (see also Baldwin et al., 2001; Coldewey-Egbers et al., 2014). Variations in ozone of up to 28 DU were found, which is
10 in very good agreement with Steinbrecht et al. (2003), who found variations of up to 25 DU using a multiple linear least-

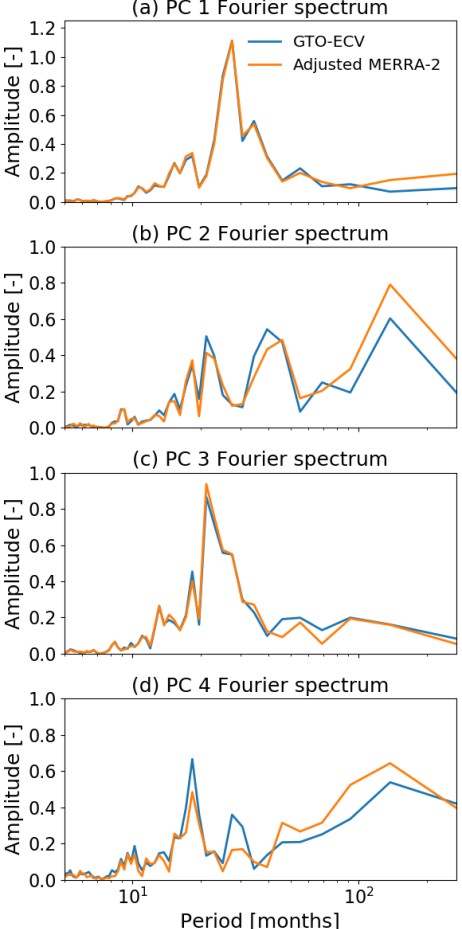

**Figure 15.** Fourier spectra of the first four principal components (shown in Fig. 14). From top to bottom: PC1, PC2, PC3, and PC4. Blue lines show GTO-ECV, and orange lines show adjusted MERRA-2.

squares analysis. For the second EOF, which was attributed to the solar cycle, a positive correlation between total ozone and the solar radio flux at 10.7 cm was detected for almost the entire tropical region. Variations up to 25 DU were found, 15 which is the same value as stated by Steinbrecht et al. (2003). EOF 3 could be attributed to a combination of two parameters, the QBO and the annual cycle, and EOF 4 could be attributed to ENSO. For the latter the maximum peak-to-peak amplitude is 10 DU, which is also in line with Steinbrecht 20 et al. (2003).

## 5  Summary and conclusions

In this paper we present a comparison of the GOME-type Total Ozone Essential Climate Variable (GTO-ECV) with the adjusted MERRA-2 (Modern Era Retrospective Analysis for 25 Research and Applications version 2) total ozone products during their 23-year overlap period from July 1995 to De-

cember 2018. The analysis is based on $5° \times 5°$ monthly mean ozone columns and associated standard deviations that are provided with the products. The main focus of this study is the assessment of the consistency among both data records concerning temporal and spatial patterns as well as interannual variability.

The GTO-ECV data record has been created in the framework of the ESA Climate Change Initiative ozone project (Coldewey-Egbers et al., 2015). It is a merged product that comprises observations from five satellite sensors (all measuring in nadir-viewing geometry, starting in 1995 with GOME/ERS-2), characterized by very high inter-sensor consistency, good spatial resolution, and near global coverage. We compare GTO-ECV with the adjusted MERRA-2 reanalysis ozone product provided by NASA. It is mainly based on the MERRA-2 data set released in 2015 (Bosilovich et al., 2015) but has been recently normalized to ozone columns from the Merged Ozone Data Set (MOD; Frith et al., 2014) in order to improve its long-term coherence.

In general, the analysis indicates a very good agreement among both data records. The mean bias between adjusted MERRA-2 and GTO-ECV monthly mean total ozone columns is $-0.9 \pm 1.5\%$. The comparison of zonally averaged data revealed that there is a small change in the behavior occurring in October 2004, when data from the Ozone Monitoring Instrument (OMI) are included in GTO-ECV and also in adjusted MERRA-2. Since the ingestion of OMI observations in both data records introduces a slight inevitable interdependence, we split the analysis into two sub-periods: July 1995–September 2004 and October 2004–December 2018. The mean difference between adjusted MERRA-2 and GTO-ECV ozone columns is $-0.5 \pm 1.8\%$ TS2 before October 2004 and $-1.0 \pm 1.3\%$ TS3 after that date. For the standard deviations the mean difference is $-0.4 \pm 3.4$ TS4 and $-1.0 \pm 1.8$ DU TS5, respectively. The small negative bias between adjusted MERRA-2 and GTO-ECV slightly increases in the later period, but the scatter in the differences is reduced. Because of the observed discontinuity in 2004, we compute the drift in the differences and found a small negative trend for the period 1995–2018 for almost all latitude bands, which is still well below 1 % per decade. The seasonal cycles agree quite well, and the differences in their amplitudes do not exceed 2 DU.

Regarding the spatial patterns of ozone and its standard deviation, both data records reveal the same general structures, though the differences indicate some minor seasonal and regional features. In the tropics differences are negative over the South Atlantic, southern Africa, and the Indian Ocean, whereas positive differences were found over the Pacific and the North Atlantic in winter and spring. Although the small overall negative bias is slightly larger in the period with OMI involved in both data records (October 2004–December 2018), the spatial pattern of the differences remains nearly the same (see Fig. 7). This might indicate that the interdependence of both data records plays only a minor role.

The variability in the differences is notably reduced in the second period (2004–2018), probably related to the enhanced data coverage and improved spatial resolution that comes along with the integration of OMI data. A similar behavior was found by Garane et al. (2018), who validated the GTO-ECV product against independent ground-based observations.

The comparison of ozone anomalies indicates an excellent agreement between both data records. For more than 97 % of the $1° \times 1°$ grid cells, the correlation coefficient is larger than 0.90. The spatial patterns of the moments of the anomalies, i.e., standard deviation and skewness, show a very good consistency. Furthermore, we assessed the interannual variability in the tropics ($25°$ N–$25°$ S) and carried out an EOF analysis. GTO-ECV and adjusted MERRA-2 exhibit a remarkable agreement in terms of spatial and temporal structures. The first four EOFs account for $\sim 92\%$ of the total variance and can be attributed to different modes of interannual climate variability. Distinct correlations with QBO, ENSO, and the solar cycle were detected.

Based on the results of our comparison, we conclude that both the GTO-ECV and the adjusted MERRA-2 total ozone data sets can be used for a number of relevant applications. GTO-ECV fulfills official user requirements (Garane et al., 2018) and is suitable for longer-term analyses that require good stability, e.g., trend studies, and for the evaluation of model simulations. The adjusted MERRA-2 was developed primarily for input into climate models and for data intercomparison studies but has not been evaluated for long-term trend studies (i.e., high spatial resolution trends) and should not be used for this purpose.

In the framework of the recently established ESA-CCI+ ozone project (http://cci.esa.int/ozone/, last access: 18 March 2020), the GTO-ECV data record will be revisited and further extended. Data from the newly launched sensors TROPOMI (Tropospheric Monitoring Instrument, launched on 13 October 2017 on board the Sentinel-5 Precursor platform) and GOME-2 on board MetOp-C (launched on 7 November 2018) will be integrated in GTO-ECV. Additionally, as part of the second phase of the European Union (EU) Copernicus Climate Change Service (C3S) ozone project GTO-ECV is regularly (every 6 months) expanded in time.

*Data availability.* The GTO-ECV Climate Research Data Package (ESA CCI, 2020) is available at http://cci.esa.int/ozone (last access: 18 March 2020) (Coldewey-Egbers et al., 2015). The adjusted MERRA-2 data record (**?**) is available via https://acd-ext.gsfc.nasa.gov/anonftp/toms/MergedOzoneData/ (last access: 18 March 2020), (Bosilovich et al., 2015).

*Author contributions.* MCE performed the analysis and wrote the paper. DL and GL initiated this comparison in the framework of the Committee on Earth Observation Satellites Atmospheric Composition Virtual Constellation (CEOS AC-VC). MCE and DL are responsible for the generation of the GTO-ECV data record. GL and SF provided the adjusted MERRA-2 ozone product. All authors contributed to the interpretation of the results and the revision of the manuscript.

*Competing interests.* The authors declare that they have no conflict of interest.

*Acknowledgements.* Melanie Coldewey-Egbers and Diego Loyola are grateful for the support by DLR, the ESA-CCI and the EU Copernicus Climate Change Service ozone projects. The SBUV Merged Ozone Data Set was constructed under the NASA Measures (Making Earth System Data Records for Use in Research Environments) Project and is maintained under NASA WBS 479717 (Long-Term Measurement of Ozone).

*Financial support.* The article processing charges for this open-access publication were covered by a Research Centre of the Helmholtz Association.

*Review statement.* This paper was edited by Alexander Kokhanovsky and reviewed by four anonymous referees.

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

## Remarks from the typesetter