# Peer review of "Comparison of GTO-ECV and Adjusted-MERRA-2 total ozone columns from the last two decades and assessment of interannual variability"

_Atmospheric Measurement Techniques, 2019_

## Referee Comment (RC1) · Anonymous Referee #4 · 27 Nov 2019

General comments

The authors present the newest version of the GTO-ECV dataset, containing long-term harmonized total-column ozone measurements from the GOME, SCIAMACHY, OMI, and GOME-2A satellites, and compare it with the Adjusted-MERRA dataset which assimilates measurements from the SBUV and SBUV-2 satellite instruments, as well as MLS, IASI, CRIS, ATMS, as well as OMI.

First, a comparison of the two datasets in terms of zonal mean ozone and its trends

and seasonal cycles is presented. Then the two datasets are analyzed and compared seasonally at 5 x 5 degree spatial resolution. Similarly, the ozone anomalies for selected 5x5 grid cells are shown and a comparison of the two datasets is made. The inter-annual variability of the anomalies is calculated as a standard deviation, and once again the two datasets are compared.

Finally, the authors perform an empirical orthogonal function (EOF) analysis on the total column ozone anomalies, within 25 degrees of the equator. The first four EOFs are found to explain 92% of the variance, and these are related to climatic indicators: quasi-biennial oscillation (QBO), solar flux, and the multivariate ENSO index (MEI).

This analysis and presentation of large public datasets is important, since users of the data will probably not do this analysis themselves before using it in their specific application. Thus, users will not be aware of potential problems which hide in the details. However, the present paper does not draw any new scientific conclusions, despite the huge effort in gathering, processing and analysis of the data.

The article is generally well written and the figures are of a suitable quality, if often quite small on the page (however, see the section on technical corrections).

Specific comments

When initially looking at Figure 1, and investigating the genealogy of the two datasets that are being compared, one's initial conclusion is that the clear change in October 2004 is because of the OMI data that is being assimilated into both of the datasets.

Therefore, it is easy to conclude that after this date, the comparison is simply OMI-to-OMI. The adjustment of the MERRA using SBUV is described in one sentence in page 5, and this subtlety is easy to miss. The start to section 3.1 on page 6, which is not very clear, further adds to the confusion.

It is almost impossible to make sense of the present paper and the data presented without a thorough reading of Coldewey-Egbers et al (2015) and Garane et al (2018), with
particular attention to the subtle changes that have been made. A genealogy/timeline of the evolution of the datasets will assist readers to understand what they are working with.

Technical corrections

Pg 1

line 11: "second period", "later period": without the context of reading further into the paper, it's not clear what this means. The abstract needs to stand alone.

Pg 3

line 33: why is the analysis limited to the low and middle latitudes? Surely total-column ozone is important at high latitudes? In your 2015 AMT paper (which I call CE2015), (pg 3924 second column, second paragraph) you state that GODFIT is robust at high SZA.

Pg 4

Line 1: "seperate" -> "separate"

Line 4: "remarkable long-term stability": where is this remarkable stability demonstrated? Perhaps you should point out here that this is different to CE2015, where GOME is used as the long-term reference. Also, in the 2015 edition of your dataset, you use a "soft-calibration" procedure, which has since been discontinued. Indeed a genealogy of the various data products (GTO-ECV and SBUV/OMI-MERRA-derived) and their versions, and the changes from version to version, would be helpful to make sense of them all.

Line 22: the url takes me to news about the ocean-colour dataset, rather than ozone.

Pg 5

Line 17: A glance at your Figure 1 shows that the discontinuities have not been removed (or perhaps they exist in the GTO-ECV dataset?). Here you speak or "renormalizing", while lower down in the page it is referred to as simply normalizing. Use consistent naming.

Pg 6

line 4: "which is completely independent of GTO-ECV": at first reading this might be taken to mean that Adjusted-MERRA is independent of GTO-ECV. Perhaps "which is itself completely..." is clearer. This paragraph as it is currently written adds to the confusion regarding the different datasets and their history.

line 17: "difference in zonal mean total ozone column is ..." here you quote a single number while talking about the zonal means. This is shown later on in the paper (i.e. Fig 5) however here it is somewhat surprising. Does this refer to the global mean?

line 27: "high latitudes and before 2002: that is probably caused by sparse data coverage..." this is in stark contrast to your detailed explanation in CE2015 where you show that the poles receive better spatial coverage.

Pg 7

Fig 1: When you say A-MERRA vs GTO-ECV, what does this mean? There is no agreed-upon meaning for 'vs' in this context. In C-E2015 you spell it out, e.g. (AM - GE) / AM. Later on in the present paper (pg 13) it becomes clear that this "vs" is not calculated how one might assume it is. Fig 1: In Garane et al 2018, their Fig 9, the lifespans of each satellite are shown as horizontal bars on a figure similar to Fig 1. It be helpful to show these, for both GTO-ECV and A-MERRA. For example, one might unkindly split the GTO-ECV into the Gome-SCIA-ECV and the OMI-ECV, such is the heavy influence of OMI measurements on the dataset; however, one might for example ponder if there has been a change since the launch of GOME-2. Might this also give a clue as to the subtle change towards the end of the time-series? Fig 1: The upper figure shows differences as a percentage, the lower figure shows differences in stdev

as Dobson units. Is this intentional?

Line 10: "differences are found the north": insert "in"

Pg 8

Table 2: Here you describe a global dataset. However, you describe DJF as being "Winter", which is only true in the northern hemisphere. Similarly for the other seasons. This is repeated several times throughout the figures and text. Line 2: "5ˆ{\circ}latitude band separately": space missing

Line 11: "introduction of OMI data into GTO-ECV data record..." OMI data is also introduced into the adjusted-MERRA dataset. This omission is made several times in the text.

Line 17 & 18: "trough" is spelled "through"

Lines 7 & 8: the seasons for a global dataset are described in terms of northern hemisphere seasons.

Page 10:

Line 1: "Atlantic Ocean, in particular in autumn" is this the North Atlantic, or the South Atlantic? Is this the boreal autumn or the austral autumn? This is extremely confusing.

Line 7 & 8: "southern hemisphere minimum ozone columns in autumn" is this the boreal or austral autumn? This is particularly confusing after reading the beginning of the paragraph, and looking at the figure.

Page 11:

Figure 4: This figure describes the seasons in terms of the northern hemisphere, for a global dataset. Also, the small title above each global map indicates the northern hemisphere seasons. Perhaps you could put the southern hemisphere season below

the map? This might make the figure too busy: you decide. This figure is very small. It could easily be split across 2 full pages.

Page 12:

Figure 5: This figure is very small. The bars are too close together in these histograms, e.g. in the left-hand pair of figures, I can't see if the blue or the orange is taller for a given total ozone amount. Are adjacent orange and blue bars meant to be for the same interval or for consecutive intervals?

Line 11: Northern-hemisphere seasons are described for a global dataset

Page 13:

Figure 6: northern hemisphere seasons in the figure and caption

Line 1: "i.e. Adj-MERRA standard deviations are higher..." if you simply give the formula for what you mean by "vs" on your graph titles (such as in CE2015), then this sort of clarification is not necessary. Indeed, this clarification makes me go back and question how I have interpreted all of your figures, since this is the opposite of my intuition.

Line 10: "corresponding seasonal cycle": is this the seasonal cycle presented in Figure 3?

Line 11: on what basis do you select your seven grid cells, or rather, the longitude at which you have selected them?

Page 14:

Line 3: "variability is dominated by the QBO". While there is clearly a biennial cycle in the data, the QBO itself is a climatological phenomenon, and you present no mechanism or evidence linking ozone column anomalies to the QBO.

Line 10: "coffcient" spelling

Page 15:

Line 4: "ozone anomalies" are these the same as the ones presented in Fig 8?

Fig 9: does [-] denote the units?

Line 5: "variability maximizes ..." this is not idiomatic English. "variability reaches a maximum..." would be better.

Line 7: "also linked to wave activity." Have you demonstrated this connection somewhere?

Line 7: "Fig 10 indicates an excellent agreement..." by using an eyeball to examine the differences on a very small plot perhaps. A plot showing the difference between (i.e. GTO-ECV - A-MERRA) the two datasets would show the agreement more clearly.

Line 11: "lower correlations between GTO-ECV and A-MERRA ozone anomalies." Are these the correlations shown in Figure 9?

Line 13: "According to Press et al..." perhaps "Following Press et al..." would be better.

Line 21: "To a large extend..." -> "extent"

Title and Line 1: Is it PCA or EOF? Please use consistent naming.

Line 7: "to a lesser extend..." -> "extent"

Line 8: Perhaps "The EOF analysis is performed on the detrended and deseasonalized 5x5 monthly mean ozone columns presented earlier" is better?

Line 9: You give a reference for the EOF analysis in line 2, do you have one describing the Savitzky-Golay filter? Why did you choose 13 months?

Fig 12: The units in on the colour-scale in the first column of plots don't match the

second column.

Line 15: "extend" -> "extent"

LIne 20: "also a positive correlation..." delete "also"

Line 8: "included in GTO-ECV." and also in A-MERRA.

LIne 13: "seaonsal"

Line 21-22: "more than 97% of the grid cells..." mention here the size of the grid cells.

---

## Referee Comment (RC2) · Anonymous Referee #2 · 2 Dec 2019

This is an extremely well-written and well-presented paper on the comparisons between space-born and modelled total columns. It is quite important for non-informed users of all the datasets described in this text to have this work as reference for their own particular applications. I strongly suggest that the authors include a clear statement on their opinion for the capabilities of these datasets: can they be used as they are for trend studies? for inter-sensor comparisons? for climate forcing applications? a paragraph with a strong message in this direction in the conclusions should be enough. Detailed comments and suggestions for improvements can be

found in the attached annotated document.

Please also note the supplement to this comment:
https://www.atmos-meas-tech-discuss.net/amt-2019-297/amt-2019-297-RC2-supplement.pdf

⎯⎯⎯⎯⎯⎯⎯⎯⎯⎯⎯⎯⎯⎯⎯⎯

---

## Referee Comment (RC3) · Anonymous Referee #3 · 11 Dec 2019

The authors present a comparison of two total ozone datasets, which generally is of interest for the atmospheric community.

Main drawback of this study is that both datasets are NOT independent, as both involve OMI measurements. This is mentioned in the paper (paper 6, line 10), but ignored in other parts and not thoroughly discussed. The GTO-ECV product, involving satellite measurements, is compared to an assimilated ozone product, also involving satellite measurements. In fact, both products involve O3 from OMI (from different algorithms, but based on the same OMI spectra). This should be clearly stated in the manuscript

(earlier than in section 3). It remains unclear to me how far the differences between GTO-ECV and MERRA after 2005 reflect just the difference between the DOAS vs. SBUV algorithm, or how far the assimilation model contributes. So please add a comparison of the OMI input data used in GTO-ECV vs. MERRA, or provide a reference on such a comparison. The impact of having data from the same instrument contributing to both datasets, and the meaning of such an intercomparison between dependent datasets, has to be discussed in more details in the manuscript.

Detailed comments:

- add a statement in the introduction that both datasets are not independent and provide arguments why the comparison still makes sense and what can be learned from it.

- Page 6 line 18: after the introduction of OMI in GTO-ECV AND in MERRA!

- Page 6 lines 22ff: when discussing differences here, the respective comparison of the input OMI data to GTO-ECV vs. MERRA has to be provided.

- Page 8 line 11: introduction of OMI in GTO-ECV AND in MERRA!

- Figure 3: please provide these plots also for before-OMI and post-OMI periods.

- Extend the discussion/conclusions wrt both datasets not being independent. What is the worth of an "excellent agreement" between two datasets that are not independent?

Minor comments:

- Table 1: please add a column for local overpass time.

- Page 4, line 13: if gridded on 1°, the smaller OMI pixels compared to SCIAMACHY do not matter that much.

- Page 5, line 18: please provide a detailed description of the "renormalization"

- Fig. 7: why do the difference plot on the right have such strong latitutde-dependency, e.g. a jump at 30°N in spring?

---

## Author Response (AR1)

*Reply to reviewer #2*

*We thank anonymous reviewer #2 for her/his valuable comments. Please find below the reviewer's comments (in black), our responses (in blue), and changes or additions to the text (in red).*

*All page / line numbers refer to the old version of the manuscript.*

*Please note that we identified an issue in the GTO-ECV data record, which affected ozone values from 2017 onward, in particular in the middle latitudes of the southern hemisphere. We had to reprocess the data record for this period. The comparison with Adjusted-MERRA was repeated and all figures were updated. In general, the main findings did not change, except for the behavior in 2017/18 in the middle latitudes of the SH (see p.6, ll.25-26), where the differences are smaller now.*

General comment:
This is an extremely well-written and well-presented paper on the comparisons between space-born and modelled total columns. It is quite important for non-informed users of all the datasets described in this text to have this work as reference for their own particular applications. I strongly suggest that the authors include a clear statement on their opinion for the capabilities of these datasets: can they be used as they are for trend studies? for inter-sensor comparisons? for climate forcing applications? a paragraph with a strong message in this direction in the conclusions should be enough.
→ We have added the following paragraph to the conclusions:
Based on the results of our comparison, we conclude that both the GTO-ECV and the Adjusted-MERRA-2 total ozone data sets can be used for a number of relevant applications. GTO-ECV fulfills official user requirements (Garane et al., 2018) and is suitable for longer-term analyses that require good stability, e.g. trend studies, and for the evaluation of model simulations. The Adjusted MERRA-2 was developed primarily for input into climate models and for data intercomparison studies, but has not been evaluated for long-term trend studies (i.e. high spatial resolution trends) and should not be used for this purpose.

Detailed comments and suggestions for improvements can be found in the attached annotated document.

p.2, l.13:
Maybe you could recommend other similar works such as:
https://rdcu.be/bQFVw
https://science.sciencemag.org/content/353/6296/269
etc.
→ We have added "Solomon et al., 2016", "Kuttippurath and Nair, 2017" and "Kuttippurath et al., 2018" to p. 2, ll. 9-10.

p.2, l.24:
Are you sure this is the most appropriate reference to make at this point?
→ We have added "Weber et al., 2018a" here.

p.2, ll.30-32:
This sentence is too vague, you can either not mention it at all or you can add proper references to the proper teams that work on the merged ozone profiles.
→ We decided to remove this sentence.

p.4, l.5:
Here you mean GOME as well as the two GOME2 sensors? Please re-write more clearly.
→ We have re-written this part of the sentence as follows:
"… while GOME, SCIAMACHY, GOME-2A, and GOME-2B are adjusted in terms of..."

An explanation of the terms "GOME-2A" and "GOME-2B" has been added to the footnote of Tab.1.

p.4, l.6:
… of the sufficiently..."
→ Corrected.

p.5, l.4:
From which time onwards?
→ We added:  "(from 1980 to September 2004)"

p.5, ll.8-10:
Shouldn't you first mention which model has actually assimilated the TOC data? This phrase reads a bit out of sequence.
→ We agree with the reviewer and added the information on the model in lines 2-5:
"It is produced with version 5.12.4 of the Goddard Earth Observing System (GEOS-5.12.4) atmospheric data assimilation system, whose key components are the GEOS-5 Atmospheric General Circulation Model (Molod et al., 2015) and the Gridpoint Statistical Interpolation (GSI) analysis scheme (Kleist et al., 2009)."

p.5, ll.10-12:
Surely you need more detail here and not just a general phrase on the "realistic" global distribution of ozone and two references. If you include more details further below in the text you could also state this here, so that it is not lacking.
→ This part now reads:
"The MERRA-2 assimilation produces realistic global distributions of ozone in the stratosphere and upper troposphere (Stajner et al., 2008; Wargan et al.,2015, Davis et al., 2017). The column ozone values agree with NASA's Total Ozone Monitoring Spectrometer (TOMS) to 1.8±2.8% in the tropics and 1.4±3.7% at higher latitudes. A more detailed validation of the MERRA-2 ozone fields and parameterized ozone chemistry are discussed in Wargan et al. (2015, 2017)."

p.5, ll.12-14:
You definitely need to include more details on the model input parameters. Which meteorology was used? which chemistry? where was the model validated? [not the ozone output, the model itself]. Where was it used in the past? and so on.
→ The MERRA-2 assimilation is based on the GEOS-5 Atmospheric General Circulation Model, which includes no input meteorology per se. The meteorology is generated by the model, which ingests temperature, pressure, and other state variables from satellite IR instruments and ground/balloon-based radiosonde measurements, among other data sources. Time dependent sea surface temperatures and sea ice data are input as boundary conditions. Only a simplified two-dimensional ozone production/loss chemistry scheme is included. Wargan et al. (2015) [https://agupubs.onlinelibrary.wiley.com/doi/full/10.1002/2014JD022493] argue that this simplified chemistry is sufficient as ozone data are being ingested daily in the assimilation process, and ozone chemical time scales throughout most of the stratosphere are longer than a day.
The validation of the MERRA-2 ozone is relevant to this publication, but a comprehensive summary of validation studies of the GEOS-5 AGCM and MERRA-2 dynamical fields is beyond the scope of this manuscript. However we include the website of the GMAO GEOS project as a source of summary information [https://gmao.gsfc.nasa.gov/GEOS_systems/].
We note that MERRA-2 is often used as meteorological/dynamical input, which, coupled to a comprehensive chemistry package, constitutes a chemistry climate model. In this work, though a model is being used, it is used in the sense of a "smart interpolater" to give high resolution spatial coverage from lower-resolution satellite data.

p.6, l.26:

...investigation…
→ Corrected.

p.8, l.2:
I know that it might make a busy graph even busier, but it might be worth adding two vertical lines at the +/-1% levels hence showing that the drift almost for all latitudes is well-within this limitation, to optically also convince the reader that these wiggles seen are not "serious". You could also of course increase the x-axis limits to +/-5%, might also achieve the same effect.
This is a simple suggestion, not massively important.
→ Thanks for the suggestion. We added two vertical lines at the +/-1% levels and increased the x-axis limits to +/-4.5%.

p.9, Reference to Figure 4:
Very nice, very useful but incredibly small plots! Either make this plot a one page plot or include columns 3&4 in a supplement and make the two first columns into one page.
→ We split the figure into two separate figures (new Figs. 4 and 5) and increased the size to enhance the readability.

p.10, ll.1-2:
You definitely need references corroborating this fact and even enumerating it at this point.
→ We expanded this point and added references. It now reads:
"Both parameters are low and nearly constant throughout the year in the tropical region, except for a little enhancement over the Atlantic Ocean. This enhancement is due to zonal variability in tropospheric ozone in terms of a persistent wave-one pattern (Fishman et al., 1992; Ziemke et al., 1996; Thompson et al., 2003), which maximizes near 0° longitude in the South Atlantic. The minimum occurs in the South Pacific near the date line. The amplitude of this wave pattern shows a seasonal variation with minimum values of ~15DU in austral autumn and maximum values of ~25DU in austral spring, associated with large-scale biomass burning in southern Africa and South America (e.g., Thompson et al., 2003).

p.11, l.6, Reference to Figure 6:
Again, this plot is very small to be able to see much, make it one page long.
→ Done. We have enlarged the figure.

p.11, l.9:
In numerics? how significant?
→ We have included numbers here. This part now reads:
Positive differences of about 0.5-1.0% occur in the tropical Pacific and in the northern part of the tropical Atlantic. On the other hand negative differences of -1.5 - -2.5% occur in the southern part of the tropical Atlantic and over southern Africa.

p.11, l.12:
Can you state some numerics from the climatologies used? this would help in quantifying if these differences are due to the climatology or whether we are seeing a "true" effect, i.e. un-related to the apriori conditions.
→ We have investigated the longitudinal structure in the tropics in more detail. The amplitude of the wave-one pattern in ozone is slightly lower for Adjusted-MERRA, which leads to the observed differences. We added:
An investigation of the zonal structure of total ozone columns from GTO-ECV and Adjusted MERRA yields that the wave-one pattern known from tropospheric columns is visible in the total column data, too. Locations of the maximum and the minimum are identical for both data records. However, the amplitude is slightly lower for Adjusted-MERRA compared to GTO-ECV, which leads to the observed longitudinal pattern in the differences (Fig. 7).

p.12, l.8:
Again, are there any numerics you can add to state how much of "non-negligible" this sampling issue can be?
→ The errors are up to ±5%. We provide this number in the text.

p.12, l.10, Reference to Figure 7:
As before, this figure is rather small.
→ We have enlarged this figure.

p.12, l.13:
Number?
→ We have added the number (2.0 – 2.5 DU).

p.13, l.1:
Number?
→ We have added the number (0.5 DU).

p.15, l.5, Reference to Figures 10 and 11:
Again, please make these slightly bigger.
→ Done. We have enlarged these figures.

p.16, l.7:
...between…
→ Corrected.

p.16, l.10:
Not sure I understand what you mean here.
→ The previous paragraph describes Figure 10, that shows the standard deviations of the ozone anomalies for two individual months (April (top) and October (bottom)). The next sentence (p.16, ll.9-11) refers to the standard deviation obtained from all months. For this, we do not show a figure. We removed "(without figure)" to avoid confusion.

p.19, l.3:
...between…
→ Corrected.

p.19, l.4:
...a result of the…
→ Corrected.

p.20, l.5:
...months…
→ Corrected.

p.20, l.5:
A beat frequency?
→ The beat frequency is the difference between two individual frequencies, which interfere. See

also p. 19, ll.5-6 for more details.

p.20, l.16:
Missing comma.
→ Corrected.

*Reply to reviewer #3*

*We thank anonymous reviewer #3 for her/his valuable comments. Please find below the reviewer's comments (in black), our responses (in blue), and changes or additions to the text (in red).*

*All page / line numbers refer to the old version of the manuscript.*

*Please note that we identified an issue in the GTO-ECV data record, which affected ozone values from 2017 onward, in particular in the middle latitudes of the southern hemisphere. We had to reprocess the data record for this period. The comparison with Adjusted-MERRA was repeated and all figures were updated. In general, the main findings did not change, except for the unclear behavior in 2017/18 in the middle latitudes of the SH (see p.6, ll.25-26), where the differences are smaller now.*

The authors present a comparison of two total ozone datasets, which generally is of interest for the atmospheric community.

Main drawback of this study is that both datasets are NOT independent, as both involve OMI measurements. This is mentioned in the paper (paper 6, line 10), but ignored in other parts and not thoroughly discussed. The GTO-ECV product, involving satellite measurements, is compared to an assimilated ozone product, also involving satellite measurements. In fact, both products involve O3 from OMI (from different algorithms, but based on the same OMI spectra). This should be clearly stated in the manuscript (earlier than in section 3). It remains unclear to me how far the differences between GTO-ECV and MERRA after 2005 reflect just the difference between the DOAS vs. SBUV algorithm, or how far the assimilation model contributes. So please add a comparison of the OMI input data used in GTO-ECV vs. MERRA, or provide a reference on such a comparison. The impact of having data from the same instrument contributing to both datasets, and the meaning of such an intercomparison between dependent datasets, has to be discussed in more details in the manuscript.

→ We agree with the reviewer that it is a little drawback of this study that both data records involve OMI measurements, which will obviously introduce an inevitable interdependence. We add a corresponding statement in the introduction. However, the data records involve OMI measurements from two different retrieval algorithms. Our opinion is that such comparison nevertheless is of value.

We broke up the analysis of the gridded data into the periods before and after the ingestion of OMI (10/2004). The spatial pattern of the differences does not change from one period to the other (see Figs. 6 and 7) which gives evidence that the differences do not reflect just the difference between the retrieval algorithms.

Furthermore, as stated in the beginning of Section 3.1 for the zonal means both data records can be regarded as virtually independent, because of the normalization of MERRA-2 w.r.t. SBUV MOD.

Detailed comments:

- add a statement in the introduction that both datasets are not independent and provide arguments why the comparison still makes sense and what can be learned from it.

→ We added a statement in the introduction (p.3, ll.13-16):

"Beginning in late 2004, total ozone column data from the OMI instrument are assimilated in the MERRA-2 reanalysis. GTO-ECV also includes OMI measurements, meaning the two data sources are not completely independent. However, the OMI data assimilated by MERRA-2 is retrieved using a different algorithm than that included in GTO-ECV. To estimate the effect of the shared OMI data on our results, we analyze differences in two periods, before and after the OMI data are included in the data products."

- Page 6 line 18: after the introduction of OMI in GTO-ECV AND in MERRA!

→ We added "and in MERRA-2" here.

- Page 6 lines 22ff: when discussing differences here, the respective comparison of the input OMI data to GTO-ECV vs. MERRA has to be provided.
→ We included basic information about both OMI ozone retrieval algorithms here, and we refer to a number of papers providing more detailed technical information, results of the geophysical validation and a comparison of both retrieval algorithms.

- Page 8 line 11: introduction of OMI in GTO-ECV AND in MERRA!
→ Here, we would like to refrain from including "and in MERRA", because for the zonal means both data records can be regarded as independent due to the normalization of MERRA-2 w.r.t. SBUV MOD (see beginning of Sec. 3.1).

- Figure 3: please provide these plots also for before-OMI and post-OMI periods.
→ We split this plot into before-OMI and post-OMI periods, but the difference is almost invisible for this kind of plot. Thus, we would prefer to leave the plot as it is.

- Extend the discussion/conclusions wrt both datasets not being independent. What is the worth of an "excellent agreement" between two datasets that are not independent?
→ We extended the summary/discussion w.r.t. both data records not being independent.

Minor comments:

- Table 1: please add a column for local overpass time.
→ Done.

- Page 4, line 13: if gridded on 1°, the smaller OMI pixels compared to SCIAMACHY do not matter that much.
→ We do not fully agree, since the smaller ground pixel size and the almost daily global coverage in case of OMI increases the representativeness of the monthly means a lot, compared to the representativeness of  monthly means obtained from SCIAMACHY data (global coverage every 6 days) alone.

- Page 5, line 18: please provide a detailed description of the "renormalization"
→ A more detailed description of the normalization is provided in lines 29-33 on the same page.

- Fig. 7: why do the difference plot on the right have such strong latitutde-dependency, e.g. a jump at 30°N in spring?
→ We think that this is related to the quite steep gradient in standard deviation which occurs at ~30°N/S, in particular in spring of the respective hemisphere. The standard deviation drops down from >40DU to <20DU within a very tight latitude range.

*Reply to reviewer #4*

*We thank anonymous referee #4 for his helpful comments and corrections. Please find below the reviewer's comments (in black), our responses (in blue), and changes or additions to the text (in red).*

*All page / line numbers refer to the old version of the manuscript.*

*Please note that we identified an issue in the GTO-ECV data record, which affected ozone values from 2017 onward, in particular in the middle latitudes of the southern hemisphere. We had to reprocess the data record for this period. The comparison with Adjusted-MERRA was repeated and all figures were updated. In general, the main findings did not change, except for the unclear behavior in 2017/18 in the middle latitudes of the SH (see p.6, ll.25-26), where the differences are smaller now.*

Anonymous Referee #4

General comments

The authors present the newest version of the GTO-ECV dataset, containing long-term harmonized total-column ozone measurements from the GOME, SCIAMACHY, OMI, and GOME-2A satellites, and compare it with the Adjusted-MERRA dataset which assimilates measurements from the SBUV and SBUV-2 satellite instruments, as well as MLS, IASI, CRIS, ATMS, as well as OMI.

First, a comparison of the two datasets in terms of zonal mean ozone and its trends and seasonal cycles is presented. Then the two datasets are analyzed and compared seasonally at 5 x 5 degree spatial resolution. Similarly, the ozone anomalies for selected 5x5 grid cells are shown and a comparison of the two datasets is made. The inter-annual variability of the anomalies is calculated as a standard deviation, and once again the two datasets are compared.

Finally, the authors perform an empirical orthogonal function (EOF) analysis on the total column ozone anomalies, within 25 degrees of the equator. The first four EOFs are found to explain 92% of the variance, and these are related to climatic indicators: quasi-biennial oscillation (QBO), solar flux, and the multivariate ENSO index (MEI). This analysis and presentation of large public datasets is important, since users of the data will probably not do this analysis themselves before using it in their specific application. Thus, users will not be aware of potential problems which hide in the details. However, the present paper does not draw any new scientific conclusions, despite the huge effort in gathering, processing and analysis of the data. The article is generally well written and the figures are of a suitable quality, if often quite small on the page (however, see the section on technical corrections).

Specific comments

When initially looking at Figure 1, and investigating the genealogy of the two datasets that are being compared, one's initial conclusion is that the clear change in October 2004 is because of the OMI data that is being assimilated into both of the datasets.

Therefore, it is easy to conclude that after this date, the comparison is simply OMI-to-OMI. The adjustment of the MERRA using SBUV is described in one sentence in page 5, and this subtlety is easy to miss. The start to section 3.1 on page 6, which is not very clear, further adds to the confusion.
→ We have expanded this part to make it more clear.

It is almost impossible to make sense of the present paper and the data presented without a thorough reading of Coldewey-Egbers et al (2015) and Garane et al (2018), with particular attention to the subtle changes that have been made. A genealogy/timeline of the evolution of the datasets will assist readers to understand what they are working with.
→ We agree with the reviewer and have reformulated the beginning of Sec. 2.1 to make it more clear.

Technical corrections

Pg 1
line 11: "second period", "later period": without the context of reading further into the paper, it's not clear what this means. The abstract needs to stand alone.
→ This part now reads:
"...whereas the difference is -1.1±1.2%  for the period from October 2004 to December 2018. The variability in the differences is considerably reduced in the period after 2004 due to..."

Pg 3
line 33: why is the analysis limited to the low and middle latitudes? Surely total-column ozone is important at high latitudes? In your 2015 AMT paper (which I call CE2015), (pg 3924 second column, second paragraph) you state that GODFIT is robust at high SZA.
→ We added:
"...middle latitudes, but also toward higher latitudes the data sets present a uniform and stable behavior."

Pg 4

Line 1: "seperate" -> "separate"
→ Corrected.

Line 4: "remarkable long-term stability": where is this remarkable stability demonstrated?
→ We now provide a reference and add:
"w.r.t. the ground-based reference (Garane et al., 2018),"

Perhaps you should point out here that this is different to CE2015, where GOME is used as the long-term reference. Also, in the 2015 edition of your dataset, you use a "soft-calibration" procedure, which has since been discontinued. Indeed a genealogy of the various data products (GTO-ECV and SBUV/OMI-MERRA-derived) and their versions, and the changes from version to version, would be helpful to make sense of them all.
→ We provide a note here on the different versions of GTO-ECV and the change of the long-term reference.

Line 22: the url takes me to news about the ocean-colour dataset, rather than ozone.
→ Replaced with the correct url.

Pg 5

Line 17: A glance at your Figure 1 shows that the discontinuities have not been removed (or perhaps they exist in the GTO-ECV dataset?). Here you speak or "renormalizing", while lower down in the page it is referred to as simply normalizing. Use consistent naming.
→ We now use "normalizing" throughout the manuscript.

Pg 6

line 4: "which is completely independent of GTO-ECV": at first reading this might be taken to mean that Adjusted-MERRA is independent of GTO-ECV. Perhaps "which is itself completely..." is clearer. This paragraph as it is currently written adds to the confusion regarding the different datasets and their history.
→ We inserted "itself" here.

line 17: "difference in zonal mean total ozone column is ..." here you quote a single number while talking about the zonal means. This is shown later on in the paper (i.e. Fig 5) however here it is somewhat surprising. Does this refer to the global mean?
→ Thanks for pointing out this inconsistency. We now state, that this is number refers to the average over all zonal means, and we provide a range for the individual zonal mean differences to make this more clear.

line 27: "high latitudes and before 2002: that is probably caused by sparse data coverage..." this is in stark contrast to your detailed explanation in CE2015 where you show that the poles receive better spatial coverage.
→ We specify this and add "...high latitudes close to the polar night and before 2002...".

Pg 7

Fig 1: When you say A-MERRA vs GTO-ECV, what does this mean? There is no agreed-upon meaning for 'vs' in this context. In C-E2015 you spell it out, e.g. (AM -GE) / AM. Later on in the present paper (pg 13) it becomes clear that this "vs" is not calculated how one might assume it is.
→ We agree and remove "A-MERRA vs. GTO-ECV" from the title and instead of that we provide the formula in the figure caption.
Fig 1: In Garane et al 2018, their Fig 9, the lifespans of each satellite are shown as horizontal bars on a figure similar to Fig 1. It be helpful to show these, for both GTO-ECV and A-MERRA. For example, one might unkindly split the GTO-ECV into the Gome-SCIA-ECV and the OMI-ECV, such is the heavy influence of OMI measurements on the dataset;
→ We can understand the reviewer's request to show horizontal bars indicating the satellites' lifespans, but we think that adding 10 lines would make the plot too confusing. For GTO-ECV the temporal coverage is provided in Table 1, and for MERRA-2 we would like to refer to Wargan et al., 2017, their Table 1.
However, one might for example ponder if there has been a change since the launch of GOME-2. Might this also give a clue as to the subtle change towards the end of the time-series?
→ The unclear change in the behavior toward the end of the time period (2017/18) has been solved, since we identified a problem in GTO-ECV, which could be eradicated. The complete analysis has been repeated and all figures were updated.
Fig 1: The upper figure shows differences as a percentage, the lower figure shows differences in stdev as Dobson units. Is this intentional?
Honestly, for us this is just a matter of taste. We prefer to show differences in standard deviation as absolute differences.

Line 10: "differences are found the north": insert "in"
→ Done.

Table 2: Here you describe a global dataset. However, you describe DJF as being "Winter", which is only true in the northern hemisphere. Similarly for the other seasons. This is repeated several times throughout the figures and text.
→ We have corrected this throughout the entire manuscript and provide either only month names or specify the hemisphere (boreal/austral). In all corresponding figures, we changed the titles.

Line 2: "5ˆ{\circ}latitude band separately": space missing
→ Corrected.

Line 11: "introduction of OMI data into GTO-ECV data record..." OMI data is also introduced into the adjusted-MERRA dataset. This omission is made several times in the text.
→ Solved.

Line 17 & 18: "trough" is spelled "through"
→ Corrected.

Lines 7 & 8: the seasons for a global dataset are described in terms of northern hemisphere seasons.
→ Solved; please see reply to comment p.8, Table 2.

Page 10:

Line 1: "Atlantic Ocean, in particular in autumn" is this the North Atlantic, or the South Atlantic? Is this the boreal autumn or the austral autumn? This is extremely confusing.
→ Solved; please see reply to comment p.8, Table 2.

Line 7 & 8: "southern hemisphere minimum ozone columns in autumn" is this the boreal or austral autumn? This is particularly confusing after reading the beginning of the paragraph, and looking at the figure.
→ Solved; please see reply to comment p.8, Table 2.

Page 11:

Figure 4: This figure describes the seasons in terms of the northern hemisphere, for a global dataset. Also, the small title above each global map indicates the northern hemisphere seasons. Perhaps you could put the southern hemisphere season below the map? This might make the figure too busy: you decide. This figure is very small. It could easily be split across 2 full pages.

→ We replaced the season in the title above each map with the names of the months that are covered. Furthermore, the figure is split into two (new Figures 4 and 5, see also reply to reviewer#2).

Page 12:

Figure 5: This figure is very small. The bars are too close together in these histograms, e.g. in the left-hand pair of figures, I can't see if the blue or the orange is taller for a given total ozone amount. Are adjacent orange and blue bars meant to be for the same interval or for consecutive intervals?
→ We increased the figure size and inserted space between the individual bars.

Line 11: Northern-hemisphere seasons are described for a global dataset
→ Solved; please see reply to comment p.8, Table 2.

Page 13:

Figure 6: northern hemisphere seasons in the figure and caption
→ Solved; please see reply to comment p.8, Table 2.

Line 1: "i.e. Adj-MERRA standard deviations are higher..." if you simply give the formula for what you mean by "vs" on your graph titles (such as in CE2015), then this sort of clarification is not necessary. Indeed, this clarification makes me go back and question how I have interpreted all of your figures, since this is the opposite of my intuition.
→ We removed "A-MERRA vs GTO-ECV" from the titles of all related figures and provide the formula in the text.

Line 10: "corresponding seasonal cycle": is this the seasonal cycle presented in Figure 3?
→ Yes, this is correct.

Line 11: on what basis do you select your seven grid cells, or rather, the longitude at which you have selected them?
→ We selected this longitude, because it covers not only the regions where the anomalies indicate an extremely high correlation, but also the region, where we found the "minimum" correlation (tropical southern Atlantic; shown later on in Fig. 9).

Page 14:

Line 3: "variability is dominated by the QBO". While there is clearly a biennial cycle in the data, the QBO itself is a climatological phenomenon, and you present no mechanism or evidence linking ozone column anomalies to the QBO.
→ We added an explanation and a reference.
"In this latitude band ozone anomalies result from a QBO-induced residual circulation, i.e. ascending/descending motion (Steinbrecht et al., 2003). For instance, westerly winds lead to downward transport and, thus, to an increase in total ozone. At the same time, less ozone-poor air from the lowermost layers is lifted upward."

Line 10: "coffcient" spelling

→ Corrected.

Page 15:

Line 4: "ozone anomalies" are these the same as the ones presented in Fig 8?
→ Yes, we computed the standard deviations from these ozone anomalies.

Fig 9: does [-] denote the units?
→ Yes.

Line 5: "variability maximizes ..." this is not idiomatic English. "variability reaches a maximum..." would be better.
→ Changed as suggested.

Line 7: "also linked to wave activity." Have you demonstrated this connection somewhere?
→ We have replaced "wave activity" with "dynamic processes" and provide two references: Hood et al. (1999) and Entzian and Peters (1999).

Line 7: "Fig 10 indicates an excellent agreement..." by using an eyeball to examine the differences on a very small plot perhaps. A plot showing the difference between (i.e. GTO-ECV - A-MERRA) the two datasets would show the agreement more clearly.
→ Instead of a plot, we would prefer to provide some numbers to underpin the good agreement. We added:
"The mean difference in the standard deviation of ozone anomalies between Adjusted-MERRA and GTO-ECV is -0.2±0.5DU or -1.5±3.8%."

Line 11: "lower correlations between GTO-ECV and A-MERRA ozone anomalies." Are these the correlations shown in Figure 9?
→ Yes. We added the reference to this Figure.

Line 13: "According to Press et al..." perhaps "Following Press et al..." would be better.
→ Changed as suggested.

Line 21: "To a large extend..." -> "extent"
→ Corrected.

Title and Line 1: Is it PCA or EOF? Please use consistent naming.
→ We now use EOF throughout the manuscript.

Line 7: "to a lesser extend..." -> "extent"

→ Corrected.

Line 8: Perhaps "The EOF analysis is performed on the detrended and deseasonalized 5x5 monthly mean ozone columns presented earlier" is better?
→ Sentence has been reformulated.

Line 9: You give a reference for the EOF analysis in line 2, do you have one describing the Savitzky-Golay filter?
→ We added a reference:
Savitzky, A. and Golay, M. J. E.: Smoothing and Differentiation of Data by Simplified Least Squares Procedures, Anal. Chem., pp. 1627–1639, https://doi.org/10.1021/ac60214a047, 1964.
Why did you choose 13 months?
→ We wanted to remove fluctuations/noise with frequencies of less than one year.

Fig 12: The units in on the colour-scale in the first column of plots don't match the second column.
→ Solved.

Line 15: "extend" -> "extent"
→ Corrected.

LIne 20: "also a positive correlation..." delete "also"
→ Done.

Line 8: "included in GTO-ECV." and also in A-MERRA.
→ Added.

LIne 13: "seaonsal"
→ Corrected.

Line 21-22: "more than 97% of the grid cells..." mention here the size of the grid cells.
→ Done.

[revised manuscript text omitted]